# Septal secretion of protein A in *Staphylococcus aureus* requires SecA and lipoteichoic acid synthesis

**Wenqi Yu, Dominique Missiakas, Olaf Schneewind***

Department of Microbiology, University of Chicago, Chicago, United States

**Abstract** Surface proteins of *Staphylococcus aureus* are secreted across septal membranes for assembly into the bacterial cross-wall. This localized secretion requires the YSIRK/GXXS motif signal peptide, however the mechanisms supporting precursor trafficking are not known. We show here that the signal peptide of staphylococcal protein A (SpA) is cleaved at the YSIRK/GXXS motif. A SpA signal peptide mutant defective for YSIRK/GXXS cleavage is also impaired for septal secretion and co-purifies with SecA, SecDF and LtaS. SecA depletion blocks precursor targeting to septal membranes, whereas deletion of *secDF* diminishes SpA secretion into the cross-wall. Depletion of LtaS blocks lipoteichoic acid synthesis and abolishes SpA precursor trafficking to septal membranes. We propose a model whereby SecA directs SpA precursors to lipoteichoic acid-rich septal membranes for YSIRK/GXXS motif cleavage and secretion into the cross-wall.
DOI: https://doi.org/10.7554/eLife.34092.001

## Introduction

Surface proteins of *Staphylococcus aureus* and other gram-positive cocci enter the secretory pathway with their N-terminal signal peptides (*DeDent et al., 2008*). Once translocated across the membrane, surface proteins are covalently linked to cell wall peptidoglycan via sortase A-catalyzed cleavage at the LPXTG motif of C-terminal sorting signals (*Schneewind et al., 1992*; *Schneewind et al., 1995*; *Mazmanian et al., 1999*). Some, but not all surface proteins are secreted at septal membranes and incorporated into cross-wall peptidoglycan (*Cole and Hahn, 1962*; *Carlsson et al., 2006*; *DeDent et al., 2008*). Following division and separation of spherical daughter cells, cross-wall anchored surface proteins are displayed over large segments of the bacterial surface (*DeDent et al., 2007*). Cross-wall trafficking of surface proteins requires a signal peptide with YSIRK/GXXS motif (*Carlsson et al., 2006*; *DeDent et al., 2008*). The YSIRK/GXXS motif is positioned N-terminal of the hydrophobic core, common to all signal peptide precursors traveling the Sec pathway (*Emr et al., 1978*; *Emr et al., 1981*; *von Heijne, 1986*).

Gram-positive bacteria rely on cell wall-anchored surface proteins for adherence to host tissues, evasion from host immune responses and acquisition of host-specific nutrients (*Foster et al., 2014*). Surface proteins with YSIRK/GXXS signal peptides are produced with high abundance and fulfill essential virulence functions during infection. For example, staphylococcal protein A (SpA) is well known for its attribute of binding to host immunoglobulin and disrupting adaptive immune responses (*Forsgren and Sjöquist, 1966*; *Kim et al., 2016*). SpA is synthesized as a precursor with an N-terminal YSIRK/GXXS signal peptide and a C-terminal LPXTG motif sorting signal (*Abrahmsén et al., 1985*; *Schneewind et al., 1992*). After initiation into the secretion pathway, the signal peptide is cleaved by signal peptidase (*Abrahmsén et al., 1985*; *Schallenberger et al., 2012*). Sortase A recognizes the LPXTG motif of the sorting signal, cleaves the polypeptide between the threonine (T) and the glycine (G) of the LPXTG motif and forms an acyl-enzyme intermediate with the C-terminal threonine (*Mazmanian et al., 1999*; *Ton-That et al., 1999*). The acyl-enzyme is

***For correspondence:**
oschnee@bsd.uchicago.edu

**Competing interests:** The authors declare that no competing interests exist.

resolved by the nucleophilic attack of the amino-group of the pentaglycine crossbridge within lipid II, the precursor for peptidoglycan synthesis (*Ton-That et al., 2000*; *Perry et al., 2002*). The product of this reaction, surface protein linked to lipid II, is then incorporated into peptidoglycan via the transglycosylation and transpeptidation reactions of cell wall synthesis (*Ton-That et al., 1997*; *Ton-That and Schneewind, 1999*).

Newly synthesized SpA is secreted into the cross-wall compartment, bounded by septal membranes of burgeoning cells during division (*DeDent et al., 2007*). Upon completion of peptidoglycan synthesis within the cross-wall, its peptidoglycan layer is split (*Frankel et al., 2011*). The adjacent cells separate and assume a spherical shape, resulting in SpA display on the bacterial surface (*DeDent et al., 2007*). Staphylococci divide perpendicular to previous cell division planes (*Tzagoloff and Novick, 1977*). By incorporating secreted polypeptides into newly synthesized cross-walls, staphylococci distribute SpA and other sortase A-anchored products over the bacterial surface (*DeDent et al., 2008*). However, not all sortase-anchored products traffic to septal membranes. Those that are secreted at polar membranes are also anchored to peptidoglycan but are not distributed over the bacterial surface (*DeDent et al., 2008*). In *S. aureus* strain Newman, thirteen different sortase-anchored surface proteins and four additional proteins are endowed with YSIRK/GXXS signal peptides for septal secretion: lipase (Lip), glycerol-ester hydrolase (Geh), murein hydrolase LytN and the cell size determinant Ebh (*Yu and Götz, 2012*; *Frankel et al., 2011*; *Cheng et al., 2014*).

The mechanisms supporting YSIRK/GXXS precursor secretion at septal membranes are not known. Here we show that the signal peptide of SpA is cleaved at the YSIRK/GXXS motif. Amino acid substitutions in the SpA signal peptide that affect cleavage at the YSIRK/GXXS motif also impair septal secretion. When used as bait for the isolation of the secretion machinery, SpA Ser$^{18}$Leu (S18L) precursor co-purified with SecA, SecDF and LtaS. We studied the contribution of these factors towards protein A secretion into the cross-wall compartment.

## Results

### SpA signal peptide variants

To facilitate the analysis of signal peptide mutants, we generated SpA$_{ED}$, a variant of protein A that is truncated for its C-terminal immunoglobulin binding domains, region X (Xr) and the LPXTG sorting signal (*Figure 1a*). *S. aureus* WY110 (Δ*spa*Δ*sbi*, pSpA$_{ED}$) cultures expressing *spa*$_{ED}$ were fractionated into culture supernatant (S) and bacterial pellet (P) and analyzed by immunoblotting. SpA$_{ED}$ was found in the extracellular medium; its precursor species was detected in the bacterial pellet (*Figure 1bc*). Site-directed mutagenesis was used to generate short deletions and amino acid substitutions in the signal peptide of SpA$_{ED}$ (*Figure 1b*). Deletion of the YSIRK motif (ΔYSIRK) diminished the abundance of the SpA$_{ED/ΔYSIRK}$ precursor and its processing (*Figure 1b*). Single amino acid substitutions at two positions in the YSIRK motif (I9S and R10A) resulted in precursor accumulation (*Figure 1b*). Further, the R10A variant exhibited diminished secretion and accumulated a precursor species that migrated faster on SDS-PAGE than the full-length precursor (*Figure 1bc*). Amino acid substitution at lysine 11 (K11A) of the YSIRK motif had no effect on SpA$_{ED/K11A}$ precursor processing and secretion (*Figure 1bc*). Deletion of GIAS (ΔGIAS) or of the two variable residues in the GXXS motif (ΔIA) caused precursor accumulation and blocked precursor processing (*Figure 1bc*). Substitution of glycine 15 (G15L) reduced the abundance of SpA$_{ED/G15L}$ and led to the accumulation of a unique precursor species that migrated faster on SDS-PAGE than full-length precursor (*Figure 1b*). Substitution of serine 18 (S18L) caused accumulation of full-length and processed precursors as well as reduced secretion (*Figure 1bc*).

*S. aureus* WY110 cultures were fractionated into culture supernatant (S), cell wall extract (W), membranes (M) and cytoplasm (C). The SpA$_{ED}$ precursor was found in the cytoplasm and membrane, whereas mature product was secreted into the culture supernatant (S) (*Figure 2a*). Precursors of the SpA$_{ED/ΔIA}$ and SpA$_{ED/R10A}$ variants accumulated mostly in the cytoplasm, whereas the SpA$_{ED/S18L}$ precursor was located predominantly in the membrane (*Figure 2ab*). Pulse-labeling experiments revealed that wild-type SpA$_{ED}$ precursor was processed within 60 s into mature, secreted product (*Figure 2c*). In contrast, processing of the SpA$_{ED/ΔIA}$, SpA$_{ED/R10A}$ and SpA$_{ED/S18L}$ precursors was delayed (*Figure 2c*). To test whether signal peptide variations affect trafficking of full-length SpA, mutations that encode the ΔIA, R10A and S18L amino acid changes were introduced into wild-type

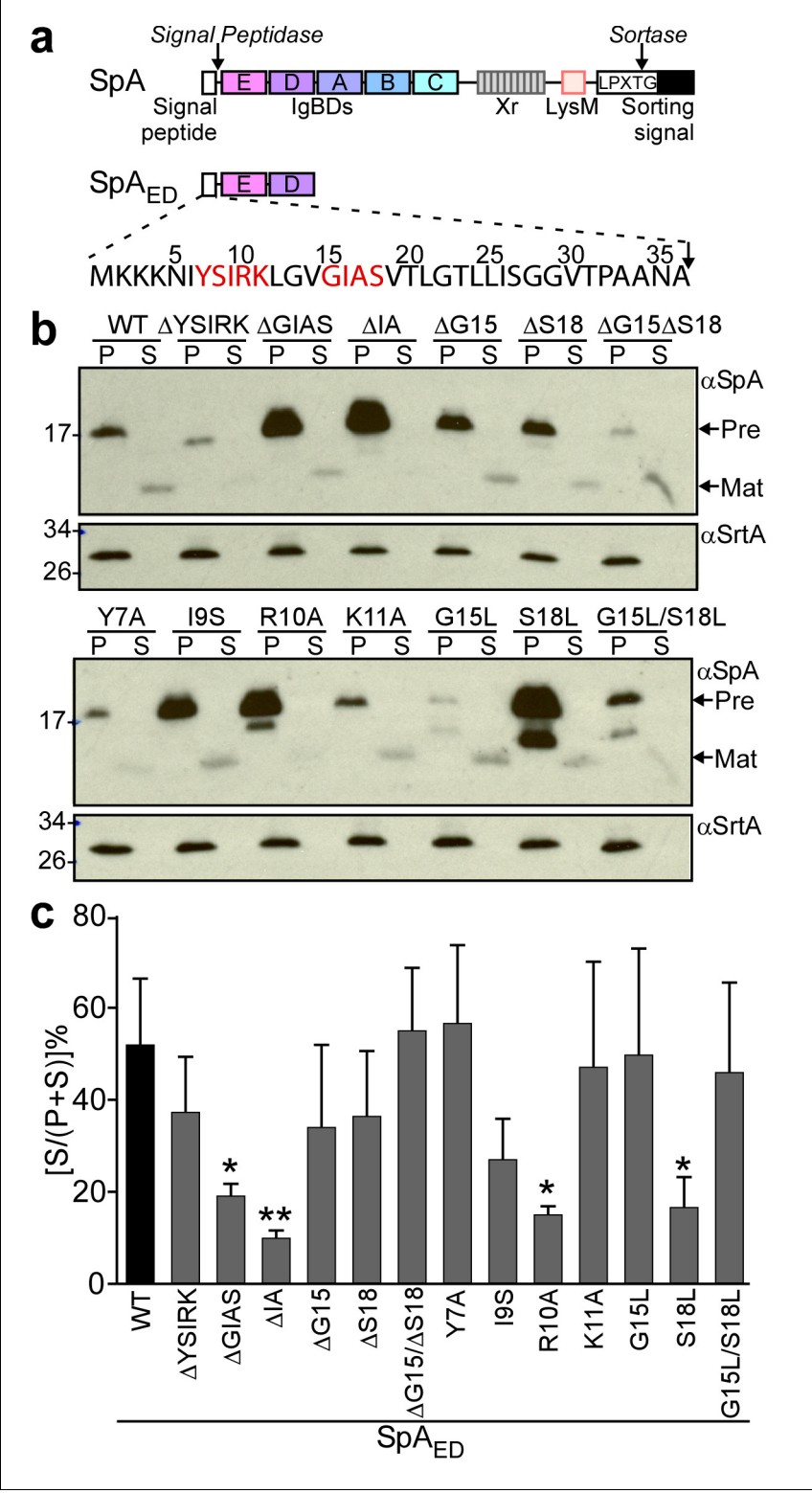

**Figure 1.** Mutagenesis of the signal peptide of staphylococcal protein A (SpA). (**a**) Schematic illustrating the primary structure of SpA and of SpA$_{ED}$ with the immunoglobulin binding domains (IgBDs, designated E, A, B, C and D), region X (Xr), LysM domain and LPXTG sorting signal. Cleavage sites for signal peptidase and sortase A are indicated. The amino acid sequence of the SpA signal peptide is displayed. YSIRK/GXXS motif residues are printed in red. (**b**) The structural genes for SpA$_{ED}$ and its variants were cloned into pOS1, expressed from the *spa* promoter in *S. aureus* WY110 (Δ*spa* Δ*sbi*) and secretion of SpA$_{ED}$ was analyzed by immunoblotting with SpA-
*Figure 1 continued on next page*

*Figure 1 continued*

specific antibody in culture supernatant (S) and lysostaphin-digested bacterial pellet (P) samples. Signal peptide bearing SpA$_{ED}$ precursors are labeled 'Pre' on the side of each blot; 'Mat' denotes mature protein without signal peptide. The calculated molecular weight (MW) of the variant precursors are: SpA$_{ED}$, 16.78 kD; SpA$_{ED/\Delta YSIRK}$,16.13 kD; SpA$_{ED/\Delta GIAS}$, 16.45 kD; SpA$_{ED/\Delta IA}$, 16.59 kD; SpA$_{ED/\Delta G15}$, 16.72 kD; SpA$_{ED/\Delta S18}$, 16.69 kD; SpA$_{ED/\Delta G15\Delta S18}$, 16.63 kD; SpA$_{ED/Y7A}$, 16.68 kD; SpA$_{ED/I9S}$, 16.75 kD; SpA$_{ED/R10A}$, 16.69 kD; SpA$_{ED/K11A}$, 16.72 kD; SpA$_{ED/G15L}$, 16.83 kD; SpA$_{ED/S18L}$, 16.8 kD; SpA$_{ED/G15L/S18L}$, 16.86 kD. The MW of SpA$_{ED}$ mature protein is 13.15 kD. Sortase A (SrtA, MW 23.54 kD) immunoblot serves as loading control. (c) Percent secretion of wild-type SpA$_{ED}$ and its variants was quantified from triplicate experiments as the intensity of immunoblotting signals in the supernatant (S) divided by the sum signals in (S + P) fractions $\times$ 100. Statistical significance was analyzed with one-way ANOVA comparing each variant with wild-type and $p$ values were recorded: WT vs. $\Delta$GIAS, p=0.031; WT vs. $\Delta$IA, p=0.0032; WT vs. R10A, p=0.0116; WT vs. S18L, p=0.0172. * denotes p<0.05, ** denotes p<0.01.

DOI: https://doi.org/10.7554/eLife.34092.002

*spa.* Wild-type and mutant staphylococci were treated with trypsin to remove all surface proteins from the bacterial surface and incubated for 20 min to allow for cell wall deposition of newly synthesized SpA. To localize SpA, bacteria were viewed by fluorescence microscopy after labeling with SpA-specific monoclonal antibody and Alexa Fluor 647-conjugated secondary IgG (red) and with BODIPY FL-vancomycin (green), which binds to cell wall peptidoglycan. As expected, wild-type SpA was assembled in the cross-wall compartment, whereas SpA$_{SP-SasF}$, a protein A variant that is secreted via a canonical (non-YSIRK/GXXS) signal peptide, was deposited into peripheral segments of the cell wall envelope (*DeDent et al., 2008*)(*Figure 2d*). SpA$_{\Delta IA}$, SpA$_{R10A}$ and SpA$_{S18L}$ exhibited defects in surface display, consistent with their observed defects in precursor processing and secretion (*Figure 2d*). Residual amounts of cross-wall localization were observed for SpA$_{R10A}$ and SpA$_{S18L}$, whereas SpA$_{\Delta IA}$ was not detected in the cross-wall compartment (*Figure 2e*). Together these data indicate that some features of the YSIRK/GXXS motif, specifically Arg[10], Ser[18] and the GXXS motif, are crucial for septal secretion of SpA in *S. aureus* (*Figure 2e*).

## Processing of SpA signal peptide variants

Wild-type SpA$_{ED}$, SpA$_{ED/\Delta IA}$, SpA$_{ED/R10A}$ and SpA$_{ED/S18L}$ were purified from staphylococcal membranes via affinity chromatography, analyzed by Coomassie-stained SDS-PAGE and identified by Edman degradation (*Figure 3a*). For wild-type SpA$_{ED}$, full-length precursor (SpA$_{ED}$-1, starting at Met[1]), as well as two precursors with faster mobility on SDS-PAGE (SpA$_{ED}$-2 and SpA$_{ED}$-3) and mature product (SpA$_{ED}$-4), that is SpsB signal peptidase-cleaved SpA$_{ED}$ starting at Ala[37], were identified (*Figure 3ab*). Edman degradation revealed that SpA$_{ED}$-2 is a product of proteolytic cleavage within the YSIRK/GXXS motif (N-terminus Gly[13]). SpA$_{ED}$-3 is a product of further cleavage, as Edman degradation identified its N-terminal amino acid 10 residues downstream (N-terminus Thr[23]) (*Figure 3ab*). Purified SpA$_{ED}$-1 precursor as well as its SpA$_{ED}$-2, SpA$_{ED}$-3 and SpA$_{ED}$-4 cleavage products were analyzed by MALDI-TOF-MS, confirming the predicted mass of the precursor and its cleaved species (*Table 1*). SDS-PAGE and Edman analysis of the SpA$_{ED/S18L}$ sample revealed the same four species as wild-type SpA$_{ED}$, albeit that the abundance of SpA$_{ED/S18L}$-1 and SpA$_{ED/S18L}$-4 was increased over that of SpA$_{ED/S18L}$-2 and SpA$_{ED/S18L}$-3 (*Figure 3ab*). Analysis of the SpA$_{ED/R10A}$ sample also identified four species, including SpA$_{ED/ R10A}$-1 precursor, SpA$_{ED/ R10A}$-3 cleavage product (N-terminus Thr[23]) and SpA$_{ED/ R10A}$-4 mature product (N-terminus Ala[37]), whereas SpA$_{ED/ R10A}$-2 represented a variant cleavage product (N-terminus Ala[10]) (*Figure 3ab*). The SpA$_{ED/\Delta IA}$ sample yielded the same precursor and cleavage species as SpA$_{ED}$ and SpA$_{ED/S18L}$, however the abundance of SpA$_{ED/\Delta IA}$-1 was increased over that of SpA$_{ED/\Delta IA}$-2, SpA$_{ED/\Delta IA}$-3 and SpA$_{ED/\Delta IA}$-4 (*Figure 3ab*). Taken together, these data indicate that the SpA precursor (SpA$_{ED}$-1) is cleaved between Leu[12] and Gly[13], which are positioned between the two motifs (underlined) of the YSIRKL/GVGIAS sequence. The R10A substitution alters the cleavage site and diminishes precursor cleavage, whereas the S18L substitution and $\Delta$IA deletion diminish precursor cleavage without altering the cleavage site between the YSIRK/GXXS motifs. Precursor cleavage between Gly[22] and Thr[23] was observed for all SpA variants, suggesting that it represents a proteolytic event unrelated to the function of the YSIRK/GXXS motif in targeting SpA secretion to septal membranes.

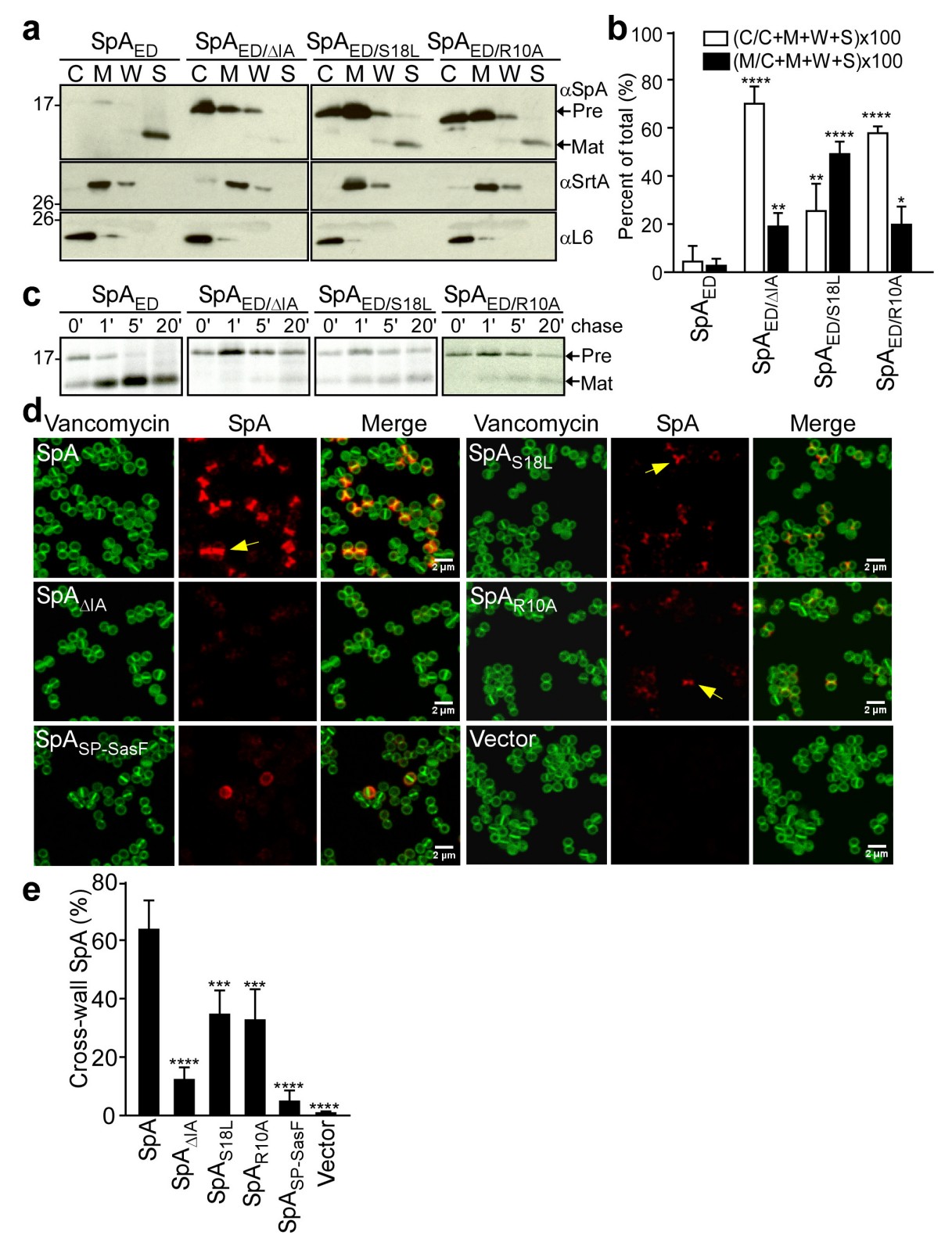

**Figure 2.** SpA signal peptide variants defective in precursor processing and septal secretion. (a) *S. aureus* cultures were fractionated into cytoplasm (C), membrane (M), cell wall (W) and culture supernatant (S) and analyzed by immunoblotting with αSpA to reveal the subcellular location of wild-type SpA$_{ED}$ precursor and secreted product of the SpA$_{ED/ΔIA}$, SpA$_{ED/S18L}$, and SpA$_{ED/R10A}$ variants. Immunoblotting with αSrtA and αL6 was used to establish fractionation and loading controls. (b) Quantification of immunoblot signal intensity in (a) using Image J. Precursor abundance (%) the bacterial

*Figure 2 continued on next page*

*Figure 2 continued*

cytoplasm (C) and membrane (M) was quantified from triplicate experiments as the intensity of immunoblotting signals divided by the sum signals in all four fractions (C + M + W + S)×100. Statistical significance was analyzed with one-way ANOVA comparing each variant with wild-type and $p$ values were recorded: for [C/(C + M + W + S)]×100, WT vs. ΔIA, p<0.0001; WT vs. S18L, p=0.0042; WT vs. R10A, p<0.0001; for [M/(C + M + W + S)]×100, WT vs. ΔIA, p=0.0056; WT vs. S18L, p<0.0001; WT vs. R10A, p=0.0405. **** denotes p<0.0001, ** denotes p<0.01, * denotes p<0.05. (c) *S. aureus* cultures were pulse-labeled for 60 s with [$^{35}$S]methionine and labeling quenched by adding an excess of non-radioactive methionine (chase). At timed intervals during the pulse (0') or 1 (1'), 5 (5'), and 20 (20') minutes after the pulse (chase), culture aliquots were precipitated with trichloroacetic acid (TCA), lysostaphin-treated, immunoprecipitated with αSpA and analyzed by autoradiography. (d) *S. aureus* WY110 (Δ*spa* Δ*sbi*) harboring chromosomal pCL55-insertions of wild-type *spa* (SpA), *spa*$_{ΔIA}$ (SpA$_{ΔIA}$) *spa*$_{S18L}$ (SpA$_{S18L}$), *spa*$_{R10A}$ (SpA$_{R10A}$) *spa*$_{SP-SasF}$ (SpA$_{SP-SasF}$) or pCL55 alone (Vector) were treated with trypsin to remove SpA. Bacteria were incubated for 20 min to allow for secretion and cell wall deposition of newly synthesized SpA. Samples were incubated with BODIPY-FL vancomycin (Vancomycin) (green) to stain the bacterial cell wall and with SpA-specific monoclonal antibody and Alexa fluor 647-labeled secondary IgG (red) to reveal SpA. (e) SpA-positive staphylococci in images derived from samples in (d) were analyzed for SpA deposition at the cross wall of diplococci (n = 200). Data from three independent experiments were used to derive the mean (± SEM) and were analyzed for significant differences with one-way ANOVA for comparisons between wild-type and mutant SpA. $p$ values were recorded: SpA vs. SpA$_{ΔIA}$, p<0.0001; SpA vs. SpA$_{S18L}$, p=0.0006; SpA vs. SpA$_{R10A}$, p=0.0004; SpA vs. SpA$_{SP-SasF}$, p<0.0001; SpA vs. Vector, p<0.0001. **** denotes p<0.0001, *** denotes p<0.001.

DOI: https://doi.org/10.7554/eLife.34092.003

## Identification of proteins that co-purify with mutant SpA precursor

We used a biochemical approach to identify staphylococcal proteins that interact with SpA precursor in septal membranes. The SpA$_{ED/S18L}$ precursor accumulates in septal membranes (*Figure 2bc*) and can be purified following solubilization with detergent (*Figure 3ab*). After crosslinking with formaldehyde, SpA$_{ED/S18L}$ precursor and associated species were isolated by affinity chromatography, heat-treated (95°C) to resolve crosslinks and analyzed by Coomassie-stained SDS-PAGE and immunoblotting with anti-SpA (*Figure 4ab*). As compared to SpA$_{ED/SP-SasF}$, several proteins co-purified with SpA$_{ED/S18L}$ and were identified by mass spectrometry (*Figure 4ab* and *Supplementary file 1*). SecA was the most abundant protein. In *E. coli*, SecA forms a homodimer and binds signal peptide-bearing precursors for subsequent translocation through the SecYEG translocon (*Grady et al., 2012*; *Tsirigotaki et al., 2017*). In agreement with this model, SecA co-purification with SpA$_{ED/S18L}$, but not SpA$_{ED}$, from staphylococcal extracts was detected by immunoblotting both in the presence and in the absence of crosslinking agent (*Figure 4c*). SecA also did not co-purify with formaldehyde-

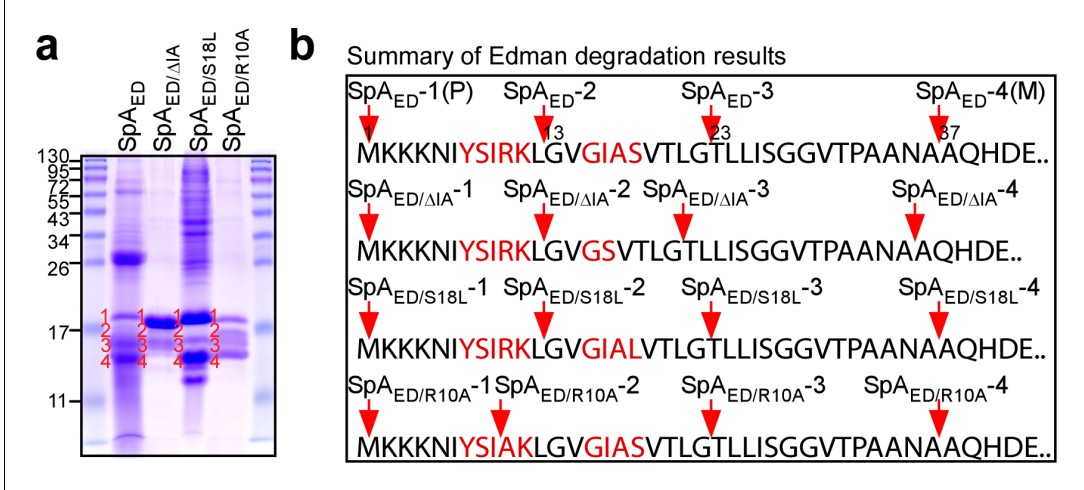

**Figure 3.** Proteolytic cleavage of the SpA signal peptide. (a) Wild-type SpA$_{ED}$ or SpA$_{ED/ΔIA}$, SpA$_{ED/S18L}$, and SpA$_{ED/R10A}$ variant precursors and cleavage products were purified from detergent-solubilized staphylococcal membranes using affinity chromatography on IgG-sepharose and analyzed on Coomassie-Blue stained SDS-PAGE. Full length precursors (1) and their cleavage products (2-4) were analyzed by Edman degradation and MALDI-TOF mass spectrometry. See *Table 1* for mass spectrometry data. (b) Schematic illustrating the proteolytic cleavage sites for each of the four precursors SpA$_{ED}$, SpA$_{ED/ΔIA}$, SpA$_{ED/S18L}$ and SpA$_{ED/R10A}$.

DOI: https://doi.org/10.7554/eLife.34092.004

**Table 1.** MALDI-TOF-MS ion signals of purified SpA$_{ED}$ species and their variants

| Protein | Observed *m/z* | Calculated *m/z*\* | Δobs.- calc. |
|---|---|---|---|
| SpA$_{ED}$-1 | 16776.47 | 16777.7 | 1.23 |
| SpA$_{ED}$-2 | 15273.92 | 15273.78 | 0.14 |
| SpA$_{ED}$-3 | 14418.92 | 14418.78 | 0.14 |
| SpA$_{ED}$-4 | 13152.40 | 13152.32 | 0.08 |
| SpA$_{ED/S18L}$-1 | 16803.02 | 16803.78 | 0.76 |
| SpA$_{ED/S18L}$-2 | 15298.89 | 15299.86 | 0.97 |
| SpA$_{ED/S18L}$-3 | 14417.89 | 14418.78 | 0.89 |
| SpA$_{ED/S18L}$-4 | 13152.22 | 13152.32 | 0.10 |
| SpA$_{ED/\Delta IA}$-1 | 16592.95 | 16593.46 | 0.51 |
| SpA$_{ED/\Delta IA}$-2 | 15088.71 | 15089.54 | 0.83 |
| SpA$_{ED/\Delta IA}$-3 | 14417.86 | 14418.78 | 0.92 |
| SpA$_{ED/\Delta IA}$-4 | 13151.55 | 13152.32 | 0.77 |
| SpA$_{ED/R10A}$-1 | 16692.99 | 16692.59 | 0.40 |
| SpA$_{ED/R10A}$-2 | 15584.97 | 15586.2 | 1.23 |
| SpA$_{ED/R10A}$-3 | 14418.46 | 14418.78 | 0.32 |
| SpA$_{ED/R10A}$-4 | 13152.07 | 13152.32 | 0.25 |

\*Based on average mass calculated with the online ExPASy tool.

DOI: https://doi.org/10.7554/eLife.34092.005

treated SpA$_{ED/SP-SasF}$ (*Figure 4d*). Most of the proteins crosslinked to SpA$_{ED/S18L}$ are components of the peptidoglycan (PBP2, MurE2, MurG, FemA, FemB, FemX), wall teichoic acid (TagB, TagF) and lipoteichoic acid synthesis pathways (LtaS) that are known to be localized to septal membranes (*Pinho and Errington, 2005*; *Mann et al., 2013*; *Reichmann et al., 2014*). We also identified EzrA, a cell division machinery component (*Steele et al., 2011*), PurL, ClpB, ClpC, PknB, as well as the products of three uncharacterized genes: SAOUHSC_01854, SAOUHSC_02423, and SAOUHSC_01583. When analyzed for SpA trafficking via immunofluorescence microscopy, *S. aureus* mutants lacking EzrA, PBP2, PurL, ClpB, ClpC, PknB, SAOUHSC_01854, SAOUHSC_02423 (UDP-*N*-acetylglucosamine pyrophosphorylase), or SAOUHSC_01583 (conserved hypothetical phage protein) did not exhibit defects in septal precursor translocation (*Figure 4—figure supplement 1*). SecA and SecDF, members of the bacterial protein secretory pathway (*Oliver and Beckwith, 1981*; *Gardel et al., 1987*; *Pogliano and Beckwith, 1994*), and LtaS, lipoteichoic acid synthase (*Gründling and Schneewind, 2007*), were selected for further study.

## SecA depletion in *S. aureus*

In *Escherichia coli*, *secA* is an essential gene (*Oliver and Beckwith, 1981*). SecA functions as an ATPase that moves many, but not all, precursor proteins across the SecYEG translocon (*Tsirigotaki et al., 2017*). To study the contribution of *secA* towards the septal secretion of SpA, we generated an inducible allele, P$_{spac}$-*secA*, in *S. aureus* WY223 (*Figure 5a*). When induced with isopropyl β-D-1-thiogalactoside (IPTG), *S. aureus* WY223 (P$_{spac}$-*secA*) forms colonies on agar and replicates in liquid media culture in a manner similar to wild-type *S. aureus* (*Figure 5bc*). However, in the absence of IPTG, *S. aureus* WY223 cannot form colonies (*Figure 5c*). Following dilution of bacteria from IPTG-containing media into broth without inducer, *S. aureus* WY223 replicates for 3 hr at a rate similar to wild-type (*Figure 5b*). Upon further dilution and incubation, *S. aureus* WY223 eventually exhibits growth retardation (6 hr time point). When analyzed by immunoblotting with SecA-specific antibody, SecA was already depleted in *S. aureus* WY223 (P$_{spac}$-*secA*) cultures 3 hr following dilution into inducer free medium (*Figure 5d*). After 6 hr of incubation, SecA could no longer be detected (*Figure 5d*). These observations indicate that depletion of SecA does not cause immediate growth retardation in *S. aureus* WY223; SecA-depletion likely impairs replenishment of translocated proteins that are required for staphylococcal growth, which manifests after 6 hr of SecA depletion.

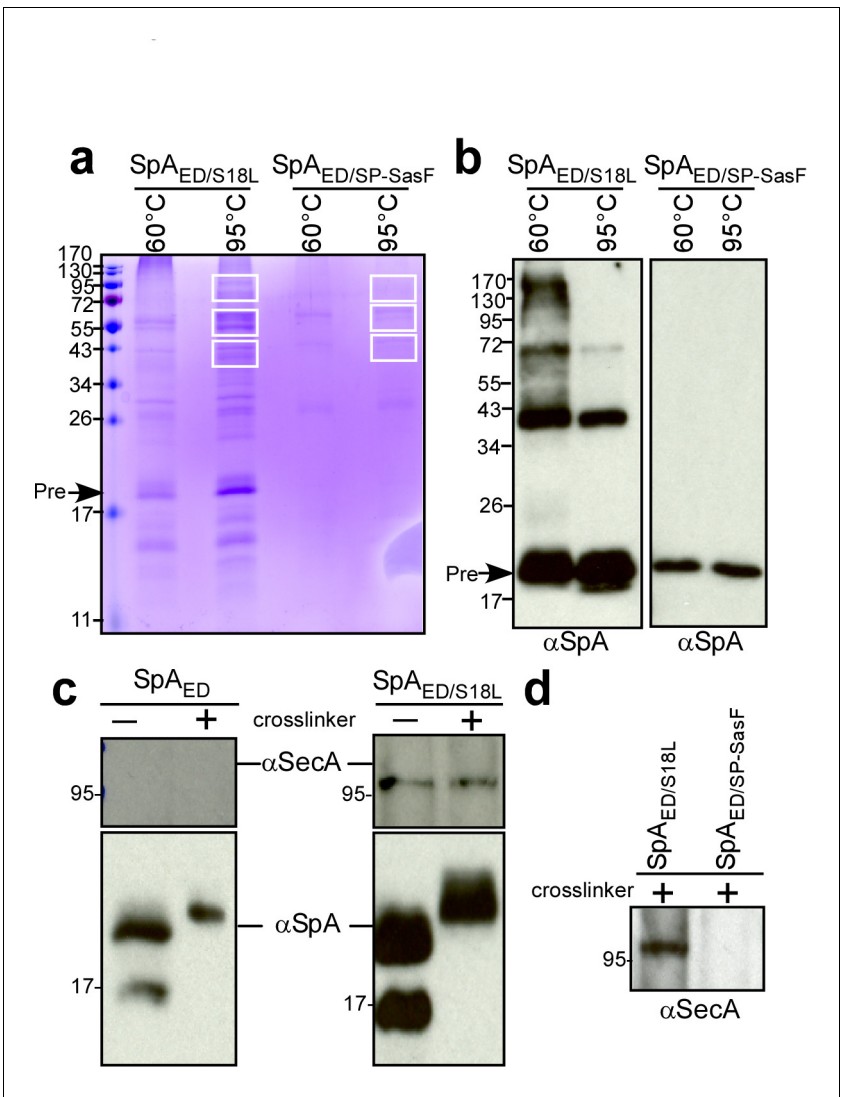

**Figure 4.** Crosslinking of staphylococcal proteins to SpA$_{ED/S18L}$ or SpA$_{ED/SP-SasF}$. (**a**) Bacteria from *S. aureus* WY110 (pSpA$_{ED/S18L}$) and *S. aureus* WY110 (pSpA$_{ED/ SP-SasF}$) cultures were crosslinked with 0.9% formaldehyde, membrane proteins detergent-solubilized and SpA$_{ED/S18L}$ as well as SpA$_{ED/SP-SasF}$ precursors purified by affinity chromatography on IgG-sepharose. Eluate was treated for 20 min at 95°C to reverse cross-linking or kept at 60°C (cross-linked control) and analyzed on Coomassie-stained SDS-PAGE. Bands were excised as indicated and individual proteins identified via ESI-MS analyses of tryptic peptides and data comparison with in silico trypsin-cleaved translation products derived from the genome sequence of *S. aureus*. Immunoblotting of 60 and 95°C samples to validate crosslinking of SpA$_{ED/S18L}$ (**b**). The identity of the SpA-immunoreactive species migrating at 43 and 72 kDa in the left panel of *Figure 4b* is not known. (**c**) Bacteria from *S. aureus* WY110 (pSpA$_{ED}$) and *S. aureus* WY110 (pSpA$_{ED/S18L}$) cultures were treated with 0.9% formaldehyde (+crosslinker) or left untreated (- crosslinker), membrane proteins detergent-solubilized, and SpA$_{ED/S18L}$ as well as SpA$_{ED}$ precursors purified by affinity chromatography on IgG-sepharose. Eluate was treated for 20 min at 95°C to reverse crosslinking and samples analyzed by immunoblotting with antibodies specific for SpA and SecA (MW 95.96 kD). (**d**) Eluates of crosslinked SpA$_{ED/S18L}$ and SpA$_{ED/SP-SasF}$ precursors were examined by immunoblotting for the presence of SecA. See ***Supplementary file 1*** for a summary of proteins crosslinked to SpA$_{ED/S18L}$.

DOI: https://doi.org/10.7554/eLife.34092.006

The following figure supplement is available for figure 4:

**Figure supplement 1.** SpA septal secretion analysis in *S. aureus* mutants with knockout or conditional mutations in genes whose products were crosslinked to SpA$_{ED/S18L}$.

DOI: https://doi.org/10.7554/eLife.34092.007

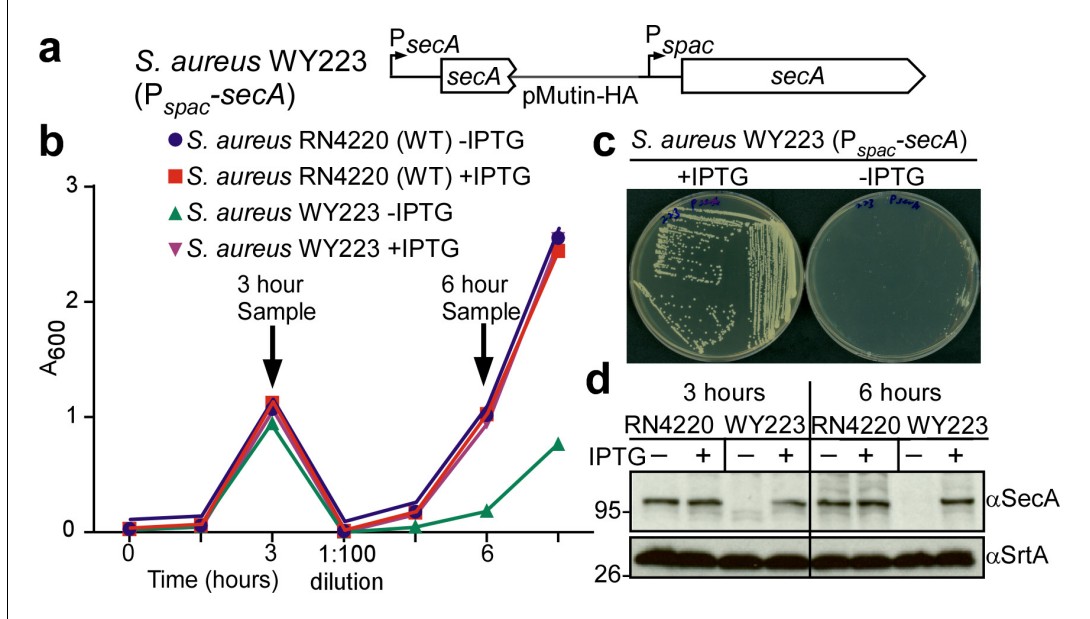

**Figure 5.** Depletion of SecA in *S. aureus*. (a) Diagram of the *secA* gene locus in *S. aureus* RN4220 (wild-type parent, WT) and its $P_{spac}$-*secA* variant. (b) Bacteria from overnight cultures of wild-type *S. aureus* and *S. aureus* $P_{spac}$-*secA* grown in TSB with 1 mM IPTG were washed and suspended in fresh TSB with or without 1 mM IPTG. Subsequent growth was monitored as increased absorbance at 600 nm ($A_{600}$). After three hours, cultures were diluted 1:100 into fresh TSB with or without 1 mM IPTG and incubated for additional growth measurements. (c) *S. aureus* $P_{spac}$-*secA* was streaked on tryptic soy agar with or without 1 mM IPTG supplement and incubated for 16 hr at 37°C for growth. (d) Culture samples retrieved after 3 and 6 hr in (b) were analyzed by immunoblotting with antibodies against SecA (αSecA) and sortase A (αSrtA).

DOI: https://doi.org/10.7554/eLife.34092.008

## SecA depletion blocks SpA secretion

After dilution into media with and without inducer, wild-type (*S. aureus* RN4220) and $P_{spac}$-*secA* (*S. aureus* WY223) were subjected to pulse labeling with [35S]methionine and protein A precursor processing was analyzed by immunoprecipitation. In wild-type, SpA precursors are processed within 60 s; similar rates of processing were observed when the $P_{spac}$-*secA* mutant was grown with IPTG inducer (*Figure 6a*). In the absence of IPTG, SpA precursor processing was slowed to about 5 min, indicating that SecA depletion inhibits precursor translocation (*Figure 6a*). When analyzed by fluorescence microcopy in trypsin-treated staphylococci incubated for 20 min without protease, wild-type *S. aureus* deposited protein A into the cross wall (*Figure 6b, yellow arrow*). Cross wall localization was diminished in $P_{spac}$-*secA* mutant bacteria grown without IPTG inducer and restored to wild-type levels when bacteria were grown in the presence of inducer (*Figure 6c*). Six hours after dilution into broth without IPTG inducer, *S. aureus* WY223 ($P_{spac}$-*secA*) cells were grossly enlarged and surrounded by a thin layer of peptidoglycan with aberrant cross-wall formation (*Figure 6d, blue arrow*); at this time point, SpA could not be detected in the bacterial envelope. As a control, growth of *S. aureus* WY223 in the presence of IPTG did not affect cell size and SpA deposition into the cell wall (*Figure 6d*).

We wondered whether SecA depletion affects the secretion of other staphylococcal proteins. Glycerol-ester hydrolase (Geh) is synthesized as a precursor with YSIRK/GXXS signal peptide motif (*Lee and Iandolo, 1986*). Following secretion at septal membranes into the cross-wall compartment, Geh is subsequently released into the extracellular medium (*Yu and Götz, 2012*). When analyzed by immunoblotting of proteins in the extracellular medium, depletion of SecA in *S. aureus* WY223 ($P_{spac}$-*secA*) caused a reduction in secreted Geh, as compared to wild-type staphylococci or *S. aureus* WY223 grown in the presence of IPTG (*Figure 6e*). Staphylococcal nuclease (Nuc), a secreted protein that contributes to the pathogenesis of human and animal infections, is synthesized as a precursor with a canonical signal peptide (*Phonimdaeng et al., 1990; Shortle, 1983*). The abundance of secreted Nuc was also diminished in SecA-depleted cultures of *S. aureus* WY223 (*Figure 6e*). As a

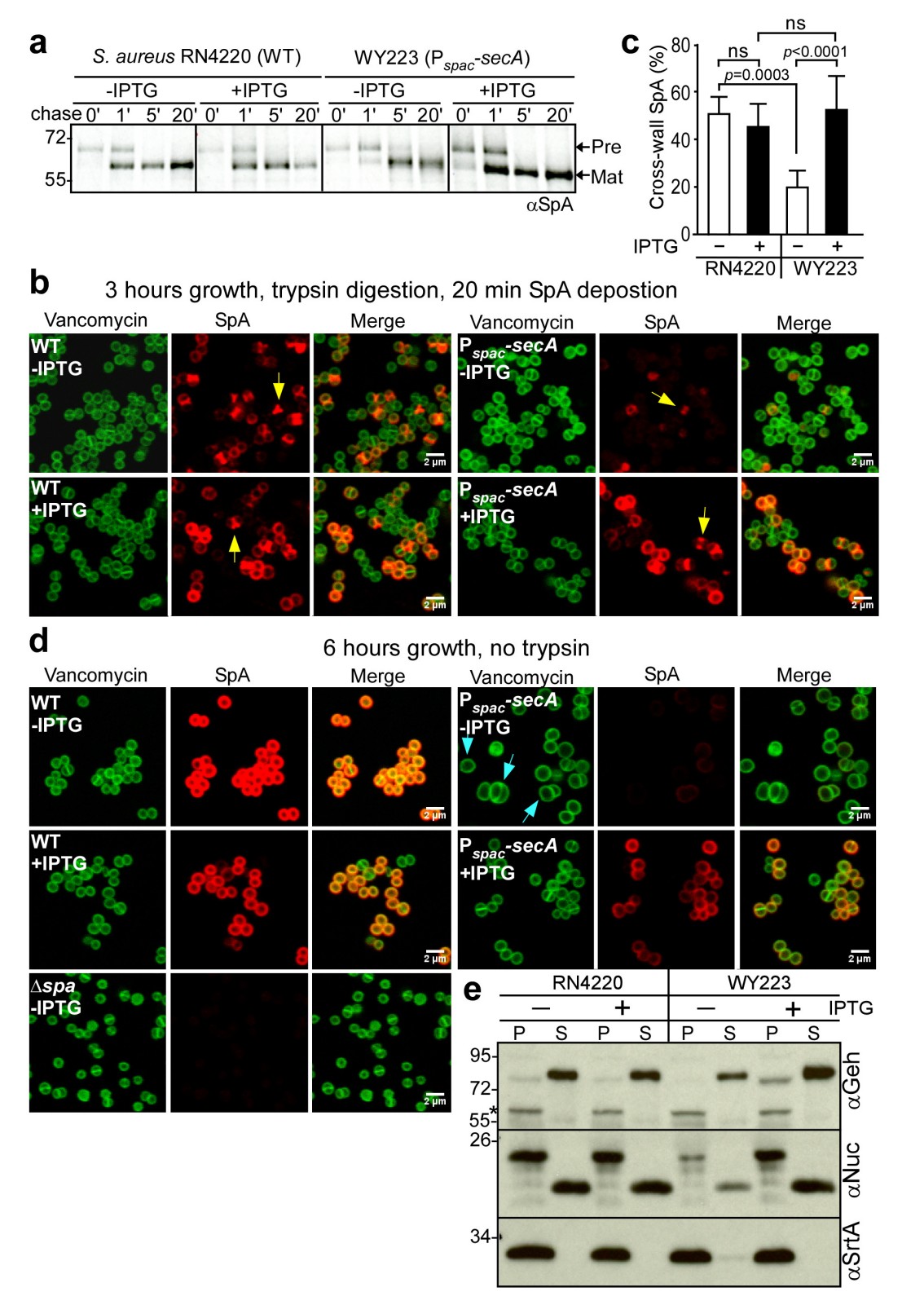

**Figure 6.** SecA depletion diminishes septal secretion of SpA in *S. aureus*. (a) SpA precursor processing of [³⁵S]methionine pulse-labeled *S. aureus* RN4220 (WT) or *S. aureus* P$_{spac}$-*secA* grown in the presence or absence of 1 mM IPTG. Bacteria were pulse-labeled for 60 s with radioactive methionine and then incubated with an excess of non-radioactive methionine. During the pulse (0′) or 1 (1′), 5 (5′) and 20 (20′) min after the addition of excess unlabeled methionine, culture aliquots were withdrawn, precipitated with TCA, digested with lysostaphin, and subjected to SDS-PAGE and

*Figure 6 continued on next page*

Figure 6 continued

autoradiography of immunoprecipitated SpA. Wild-type *S. aureus* (WT) and its P*spac*-secA variant were grown for 3 (**b**) and 6 hr (**d**) in the presence or absence of 1 mM IPTG (see *Figure 5*). Samples in (**b**) were treated with trypsin to remove SpA from the bacterial surface. Bacteria were incubated for 20 min to allow for secretion and cell wall deposition of newly synthesized SpA. Samples were incubated with BODIPY-FL vancomycin (green) to stain the bacterial cell wall and with SpA-specific monoclonal antibody and Alexa Fluor 647-labeled secondary IgG (red) to reveal SpA. As a control for SpA-specific staining, the *S. aureus* Δspa variant grown in the absence of IPTG was analyzed by fluorescence microscopy. (**c**) SpA-positive staphylococci in images derived from samples in (**b**) were analyzed for SpA deposition at the cross wall of diplococci (n = 200). Data from three independent experiments were used to derive the mean (± SEM) and were analyzed for significant differences with one-way ANOVA and $p$ values recorded: RN4220-IPTG vs. RN4220 +IPTG, non-significant (ns); RN4220-IPTG vs. WY223-IPTG, p=0.0003; WY223-IPTG vs. WY223 +IPTG, p<0.0001, RN4220 +IPTG vs. WY223 +IPTG, ns. (**e**) SecA depletion diminishes secretion of staphylococcal proteins. Protein samples from the extracellular supernatant (S) and bacterial pellet (P) of *S. aureus* RN4220 (WT) and *S. aureus* P*spac*-secA cultures grown for 3 hr in the presence or absence of 1 mM IPTG were analyzed by immunoblotting with antibodies against glycerol-ester hydrolase (αGeh) (precursor MW 76.39 kD, mature protein MW 72.26 kD), nuclease (αNuc) (precursor MW 25.12 kD, mature protein MW 18.78 kD) and sortase A (αSrtA). *Identifies unknown proteins crossreactive with αGeh.

DOI: https://doi.org/10.7554/eLife.34092.009

control, production of sortase A in staphylococcal membranes was not affected by the depletion of SecA. Taken together, these data indicate that SecA is essential for *S. aureus* growth and for the secretion of precursors with canonical and YSIRK/GXXS signal peptides.

## Localization of SecA and SpA precursors in staphylococci

To localize SecA within *S. aureus*, we generated a translational hybrid between *secA* and the structural gene for super-folder green fluorescent protein (*gfp*) (*Pédelacq et al., 2006*) under transcriptional control of the P*tet* promoter in *S. aureus* WY230 (P*spac*-secA, P*tet*-secA:sfGFP, *Figure 7a*). Expression of *secA:sfGFP* in the P*spac*-secA variant restored bacterial growth in the absence of IPTG inducer, indicating that *secA-gfp* is functional (*Figure 7b*). Growth restoration occurred in the presence and in the absence of anhydrotetracycline (ATc), suggesting that *secA:sfGFP* must be expressed even in the absence of the P*tet* inducer (*Figure 7b*). Immunoblotting of staphylococcal cell extracts 3 and 6 hr following dilution into media lacking IPTG revealed that *S. aureus* WY230 indeed produced small amounts SecA:sfGFP in the absence of ATc (*Figure 7c*). In the presence of ATc inducer, the abundance of SecA-GFP was increased (*Figure 7c*). As expected, wild-type SecA was depleted when *S. aureus* WY230 was cultured for 3 or 6 hr without the IPTG inducer (*Figure 7b*). However, under SecA depleting conditions, even small amounts of SecA-sfGFP in *S. aureus* WY230 (-ATc) restored precursor processing of pulse-labeled SpA (*Figure 7d*). SpA precursor processing was accelerated to levels faster than wild-type when *S. aureus* WY230 cultures were grown in the presence of ATc (*Figure 7d*). Fluorescence microscopy of *S. aureus* WY230 stained with the membrane dye FM4-64 (red) revealed SecA-sfGFP localization to plasma membranes (*Figure 7e*). In dividing cells, SecA-sfGFP was found on septal and on polar membranes (*Figure 7e*). ATc-induced overexpression of SecA-sfGFP caused accumulation of hybrid protein throughout plasma membranes (*Figure 7e*). Thus, in *S. aureus* WY230, SecA-sfGFP is associated with all membranes and is not restricted to septal membranes.

The envelope of trypsin-treated, paraformaldehyde-fixed *S. aureus* WY223 (P*spac*-secA) was permeabilized with murein hydrolase and with detergent to detect intracellular precursors via microscopy with fluorescent antibody (*Harry et al., 1995*; *Pinho and Errington, 2003*). In *S. aureus* WY223 producing wild-type levels of SecA (P*spac*-secA +IPTG), SpA precursors were localized to septal membranes (*Figure 8a*). In other images, SpA precursors appeared as two puncta or ring deposits at septal membranes, reminiscent of FtsZ and of the division rings that are known to accumulate at this site (*Figure 8ab*) (*Lutkenhaus, 1993*). In contrast, under SecA depleting conditions (-IPTG), SpA precursors in *S. aureus* WY223 were associated with polar membranes and were not localized to septal membranes (*Figure 8a*). These results suggest that in staphylococci with a functional secretion pathway, SpA precursor are localized to the vicinity of septal division rings. However, in cells lacking functional secretion machines, SpA precursors are located throughout the cytoplasm and cannot traffic to septal membranes.

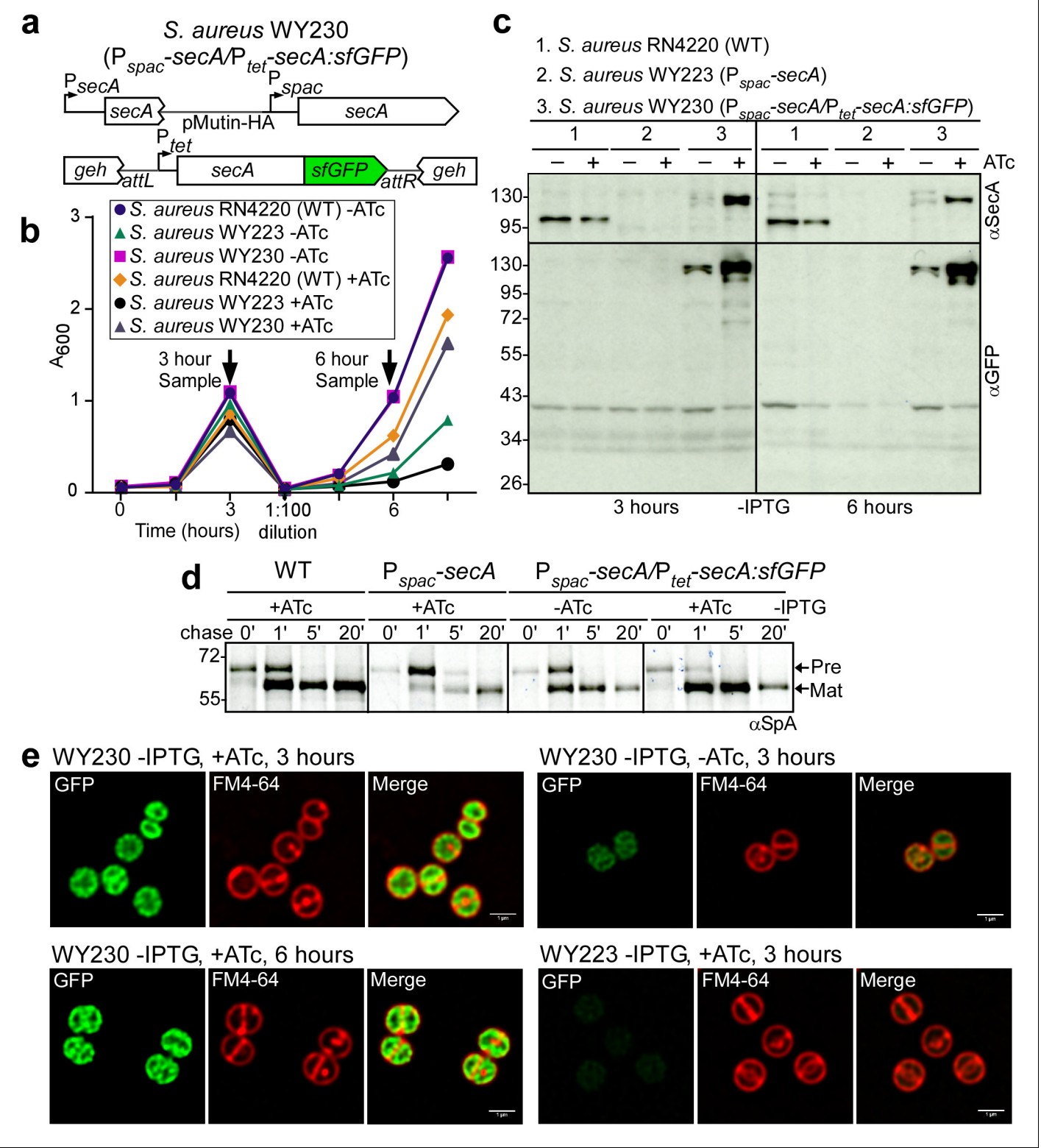

**Figure 7.** SecA localization in staphylococci. (a) Diagram of the *secA* gene locus and of the pCL55-mediated *att* insertion site for *secA-sfGFP* in the staphylococcal genome. (b) Bacteria from overnight cultures of *S. aureus* RN4220 (WT), *S. aureus* P$_{spac}$-*secA* (WY223) and *S. aureus* P$_{spac}$-s*ecA*/P$_{tet}$-*secA: sfGFP* (WY230) grown in TSB with 1 mM IPTG were washed and suspended in fresh TSB without IPTG and with or without 0.43 µM anhydro-tetracycline (ATc); growth was monitored as increased absorbance at 600 nm (A$_{600}$). After three hours, cultures were diluted 1:100 into fresh TSB without IPTG and with or without 0.43 µM ATc and incubated for further growth measurements. (c) Culture samples retrieved after 3 and 6 hr from the experiment
*Figure 7 continued on next page*

*Figure 7 continued*

detailed in (**b**) were analyzed by immunoblotting with rabbit antibodies against SecA (αSecA) and sfGFP (αGFP). (**d**) [$^{35}$S]methionine-labeled *S. aureus* cultures incubated for 3 hr as described in (**b**) were analyzed during the 60 s pulse with radioactive methionine (0) and 1, 5 and 20 min after the addition of excess unlabeled methionine via SDS-PAGE and autoradiography of immunoprecipitated SpA. (**e**) Fluorescence microscopy of bacteria from *S. aureus* cultures incubated for 3 and 6 hr as described in (**b**). Bacteria were stained with the membrane dye FM4-64 (red) and analyzed for SecA-sfGFP fluorescence (green). Scale bar, 1 μm.

DOI: https://doi.org/10.7554/eLife.34092.010

## SecDF contributes to SpA secretion

The *secDF* gene is not essential for protein secretion and *S. aureus* growth, however *secDF* mutants exhibit diminished secretion of many precursors secreted via canonical and YSIRK-GXXS signal peptides (*Quiblier et al., 2011*; *Quiblier et al., 2013*). SecDF is a member of the resistance nodulation and cell division (RND) membrane protein family with 12-transmembrane spanning segments. SecDF functions as a membrane-integrated chaperone. Sustained by the proton motive force, SecDF catalyzes ATP-independent translocation and folding of proteins on the *trans*-side of the plasma membrane (*Tsukazaki et al., 2011*). *S. aureus* expresses two additional RND proteins, here designated Rnd2 (SAOUHSC_02525) and Rnd3 (SAOUHSC_02866) (*Quiblier et al., 2011*). The *rnd2* gene is located downstream of *femX*, whose product tethers glycine from glycyl-tRNA to the ε-amino group of lysine in lipid II peptidoglycan precursor [$C_{55}$-$(PO_4)_2$-MurNAc(L-Ala-D-iGln-L-Lys-Da-Ala-D-Ala)-GlcNac] (*Rohrer et al., 1999*). Rnd2 product interacts with FemB, PBP1 and PBP2 (*Quiblier et al., 2011*). As SecDF, FemB and PBP2 were each found crosslinked to SpA$_{ED/S18L}$ precursors (*Supplementary file 1*), we asked whether *secDF*, *rnd2* and *rnd3* contribute to septal secretion of SpA. Compared to wild-type *S. aureus*, the Δ*secDF* mutant (*S. aureus* WY418) accumulated SpA$_{ED}$ precursor in bacterial cells and secreted reduced amounts of mature SpA$_{ED}$ into the extracellular medium (*Figure 9ab*). *S. aureus rnd2* (WY416) and *rnd3* (WY400) mutants exhibited wild-type levels of SpA$_{ED}$ secretion (*Figure 9ab*). A variant (WY412) lacking all three genes, Δ*secDF* Δ*rnd23*, accumulated precursors at a level similar to the Δ*secDF* mutant (*Figure 9ab*). When analyzed for other secreted proteins, the Δ*secDF* mutant secreted diminished amounts of Geh and failed to secrete Coa, whose precursor is secreted via a canonical signal peptide (*Phonimdaeng et al., 1990*), while the Δ*rnd2* and Δ*rnd3* variants displayed wild-type phenotypes (*Figure 9ab*). Thus, SecDF, but not Rnd2 and Rnd3, contributes to protein secretion in *S. aureus*. Immunofluorescence microscopy experiments revealed that septal secretion of SpA was diminished in the Δ*secDF* and Δ*secDF* Δ*rnd23* mutants (*Figure 9c*). Unlike SecA-depleted cells, where SpA precursors failed to associate with septal membranes, Δ*secDF* and Δ*secDF* Δ*rnd23* mutants exhibited puncta of SpA precursors and rings of low fluorescent intensity at septal membranes (*Figure 9c*). The diminished abundance of SpA precursors at septal membranes was restored to wild-type levels by plasmid-borne expression of *secDF* in Δ*secDF* (pSecDF) and in Δ*secDF* Δ*rnd23* (pSecDF) staphylococci (*Figure 9c*). These data suggest that SecDF aids in the translocation of SpA across staphylococcal membranes but is not absolutely required for precursor targeting to septal membranes.

## LtaS is required for septal localization of SpA

LtaS-mediated synthesis of lipoteichoic acid, a polyglycerol-phosphate polymer decorated with esterified D-alanyl and GlcNAc residues, is essential for *S. aureus* growth and cell division (*Gründling and Schneewind, 2007*). Earlier work generated *S. aureus* P$_{spac}$-*ltaS*, a strain with IPTG-inducible expression of lipoteichoic acid synthase (*Figure 10a*). In the absence of IPTG inducer, LtaS is depleted in *S. aureus* ANG499 (P$_{spac}$-*ltaS*), providing an experimental system to study the effects of LTA synthesis on septal secretion of SpA (*Gründling and Schneewind, 2007*). Surface proteins were removed with trypsin and staphylococci were incubated for 20 min to localize deposition of newly synthesized SpA (*Figure 10bc*). LtaS depletion (-IPTG) resulted in SpA deposition into polar peptidoglycan, whereas under LtaS inducing conditions (+IPTG) SpA was localized in the cross-wall (*Figure 10bc*). We asked whether SpA precursors are mislocalized to polar membranes under conditions of LtaS depletion. Microscopic analysis of trypsin-treated, lysostaphin- and detergent-permeabilized staphylococci revealed SpA targeting to septal membranes in wild-type (*S. aureus* RN4220) and in IPTG-induced *S. aureus* ANG499 (P$_{spac}$-*ltaS*)(*Figure 10d*). In contrast, without IPTG inducer, *S.*

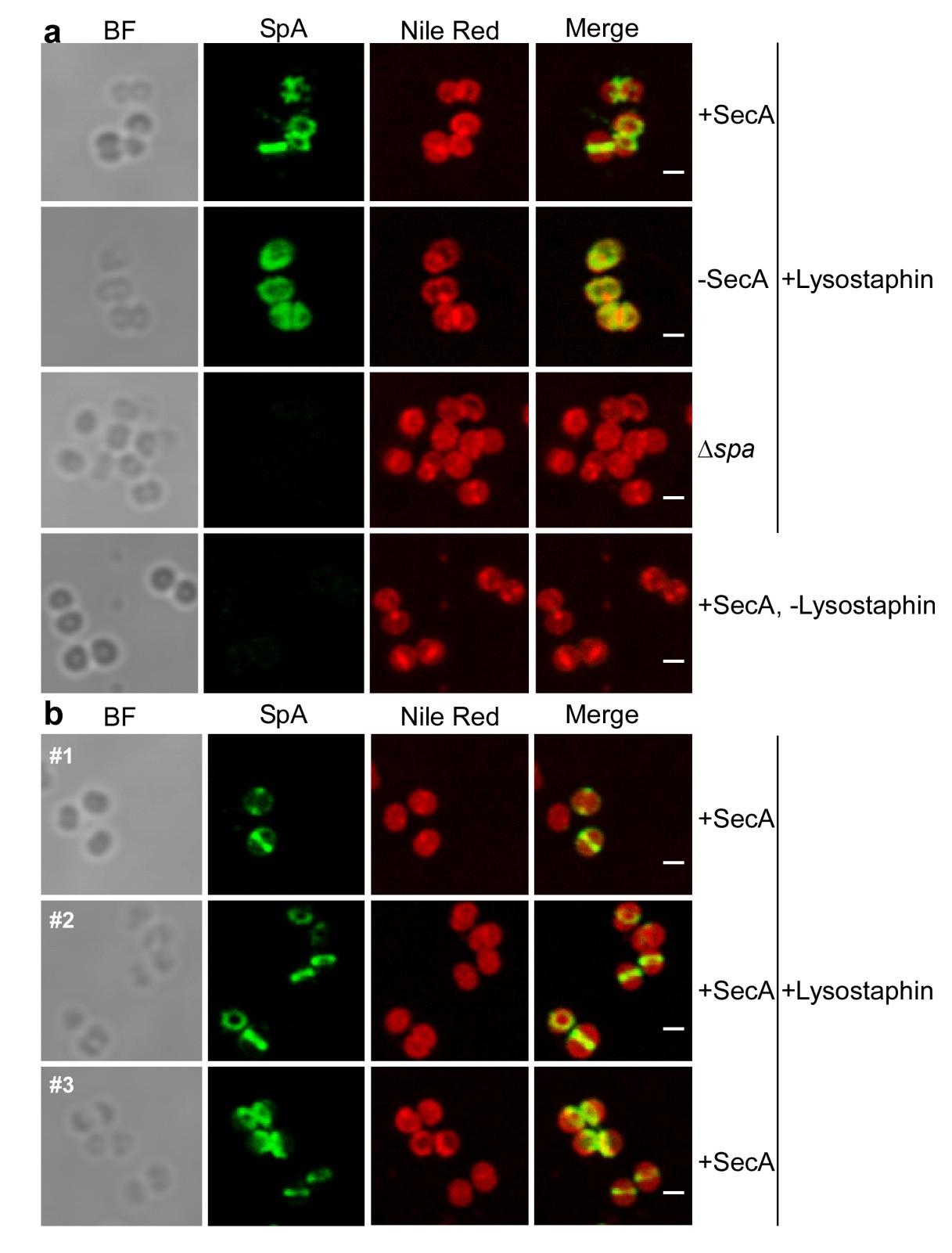

**Figure 8.** Intracellular trafficking of SpA in the presence and absence of SecA. (**a**) *S. aureus* P$_{spac}$-*secA* cells were grown for 3 hr in the presence (+SecA) or absence of 1 mM IPTG (-SecA) and, alongside *S. aureus* Δ*spa* control cells, were trypsin treated to remove extracellular surface proteins and fixed with para-formaldehyde. Samples were then treated with lysostaphin (+Lysostaphin) or left untreated (-Lysostaphin), incubated with detergent and SpA-specific rabbit antibodies and Alexa Fluor 488-labeled goat-anti-rabbit-IgG (green) and with Nile red to reveal bacterial membranes. Bright-field

*Figure 8 continued on next page*

Figure 8 continued

microscopy (BF) images were acquired to reveal the contours of all bacterial cells. Scale bar, 1 μm. (b) Additional samples (#1, #2 and #3) of *S. aureus* P*spac*-*secA* cells were grown in the presence of 1 mM IPTG (+SecA), trypsin treated, fixed with para-formaldehyde, lysostaphin treated, incubated with detergent and with SpA-specific antibody (green) and Nile red.
DOI: https://doi.org/10.7554/eLife.34092.011

*aureus* ANG499 mislocalized SpA precursors to polar membranes (*Figure 10d*). Protein secretion and cell wall anchoring of SpA were analyzed by immunoblotting in *S. aureus* cultures separated into culture supernatant (S) and bacterial sediment (P, pellet) samples. These experiments revealed that LtaS depletion in *S. aureus* ANG499 (P*spac*-*ltaS*, -IPTG) diminished the abundance of cell wall anchored SpA without affecting the secretion of Geh and Nuc (*Figure 10e*). Consistent with the immunoblotting results, LtaS depletion diminished the overall surface distribution of SpA (*Figure 10f*), in agreement with the hypothesis that cross-wall targeting via the YSIRK/GXXS signal peptide, but not polar secretion, is responsible for efficient surface distribution of proteins in staphylococci (*Carlsson et al., 2006*; *DeDent et al., 2008*). Together these data indicate that LtaS depletion and a block in lipoteichoic acid synthesis abolished SpA precursor trafficking to septal membranes without affecting its secretion at polar membranes.

## Discussion

Cell wall-anchored surface proteins with YSIRK/GXXS motif signal peptides have been identified in streptococcal and staphylococcal species (*Tettelin et al., 2005*; *Rosenstein and Götz, 2000*). Although sortase-anchored surface proteins are found in many different gram-positive bacteria, the signal peptides of surface proteins in the genus *Actinomyces*, *Bacillus*, *Clostridium*, and *Listeria* do not contain the YSIRK/GXXS motif. Common features of staphylococci and streptococci are their spherical or ovoid cell shapes and cell wall synthesis programs at septal membranes; in staphylococci this compartment is designated as the cross-wall (*Giesbrecht et al., 1976*; *Touhami et al., 2004*; *Monteiro et al., 2015*). Earlier work demonstrated that the YSIRK/GXXS motif of the SpA precursor is dispensable for sortase-catalyzed cell wall anchoring (*Bae and Schneewind, 2003*). However, precursors with YSIRK/GXXS motif signal peptides are targeted for secretion at septal membranes and sortase-mediated deposition into the cross wall compartment (*Carlsson et al., 2006*; *DeDent et al., 2008*). After completion of cross-wall synthesis, peptidoglycan splitting and cell separation, the anchored products of *spa* and of other genes with YSIRK/GXXS motif signal peptides are distributed over the bacterial surface (*Cole and Hahn, 1962*; *DeDent et al., 2007*). In contrast, surface proteins with canonical signal peptides are deposited by sortase into polar peptidoglycan but cannot be distributed over bacterial surfaces (*Carlsson et al., 2006*; *DeDent et al., 2008*).

Although it is clear that YSIRK/GXXS signal peptides are necessary and sufficient for septal secretion of proteins, the mechanisms supporting such trafficking were heretofore not known. We show that the YSIRK/GXXS signal peptide of SpA is cleaved between Leu[12] and Gly[13], separating the YSIRK sequence from the GXXS motif and from the remainder of the signal peptide. Amino acid substitutions and deletions that affect precursor cleavage and secretion map to three of the four residues (underlined) that are strictly conserved in the YS<u>IR</u>K/<u>G</u>XX<u>S</u> motif: Ile[9], Arg[10] and Ser[18]. Substitution of the fourth residue, Gly[15], with Leu resulted in diminished abundance of precursor substrate and secreted product and in accumulation of a cleaved precursor species (*Figure 1b*). On the basis of these observations, we are compelled to speculate that YSIRK/GXXS motif cleavage may represent a mechanism for precursor translocation at septal membranes. For example, the YSIRK/GXXS motif may inhibit a key function of the adjacent hydrophobic core within signal peptides: promoting the membrane translocation of precursors. Such inhibitory mechanism could be relieved by a YSIRK/GXXS protease that localizes to the septal membrane. Other mechanisms of proteolytic control for YSIRK/GXXS mediated signal peptide function can also be thought of. Importantly, the discovery of two sequential proteolytic events, YSIRK/GXXS motif cleavage and the signal peptidase-mediated cut provide experimental opportunities for the testing of predictive models. SpA precursors were also cut between Gly[22] and Thr[23], a site that is located within the hydrophobic core of the signal peptide. Mass spectrometry analysis of the *S. aureus* COL secretome also identified signal peptide fragments that had been generated by cleavage in the hydrophobic core, including SpA

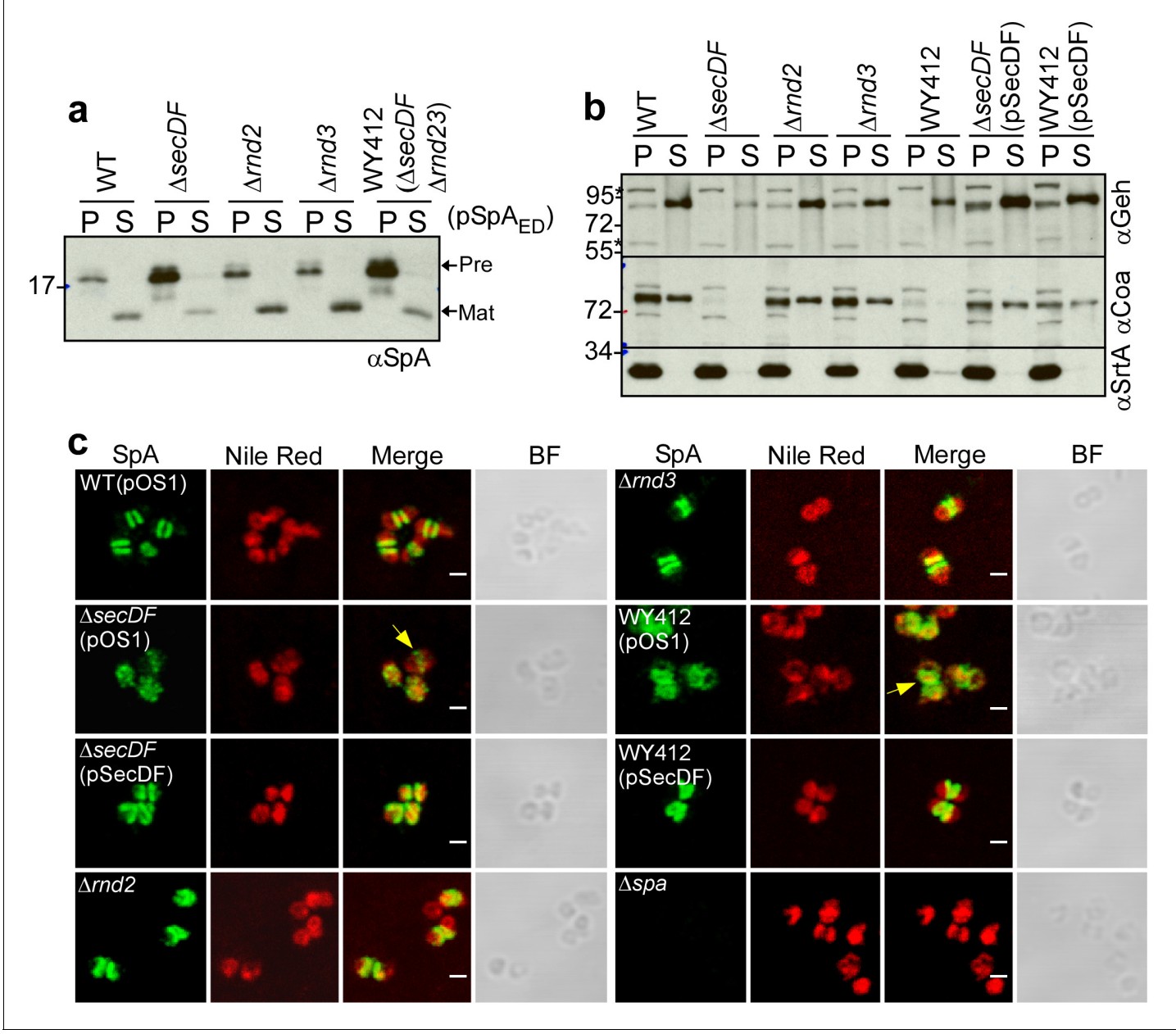

**Figure 9.** SecDF contributes to SpA secretion at septal membranes. (a) *S. aureus* cultures were centrifuged to sediment the bacteria into the pellet (P) fraction and separate them from the extracellular medium (S, supernatant). Following lysostaphin digestion of bacteria, proteins in both fractions were precipitated with TCA and analyzed by immunoblotting with αSpA. (b) *S. aureus* cultures were fractionated as described in (a) and subjected to immunoblotting with antibodies specific for glycerol-ester hydrolase (αGeh), coagulase (αCoa) (precursor MW 71.72 kD, mature protein MW 68.96 kD) and sortase A (αSrtA). (c) Fluorescence microscopy of bacteria from cultures of *S. aureus* RN4220 (WT, wild-type), WY418 (Δ*secDF*), WY416 (Δ*rnd2*), WY400 (Δ*rnd3*) and WY412 (Δ*secDF* Δ*rnd23*) mutants with and without expression plasmid for wild-type *secDF* (pSecDF) as well as *S. aureus* SEJ1 (Δ*spa*) as control. Bacteria were trypsin treated to remove extracellular surface proteins and fixed with para-formaldehyde. Samples were treated with lysostaphin, incubated with detergent and SpA-specific rabbit antibodies and Alexa Fluor 488-labeled goat-anti-rabbit-IgG (green) and with Nile red to reveal bacterial membranes. BF identifies the bright-field microscopy view of fluorescence microscopy images. Scale bar, 1 μm.
DOI: https://doi.org/10.7554/eLife.34092.012

signal peptides cleaved between Gly$^{22}$ and Thr$^{23}$ (*Ravipaty and Reilly, 2010*). The significance of signal peptide cleavage in the hydrophobic core is not known, as amino acid substitutions preventing such proteolysis have not be studied for their effect on protein secretion or membrane integrity. We presume that cleavage at Gly$^{22}$/Thr$^{23}$ may not be related to septal secretion. Cleavage at the

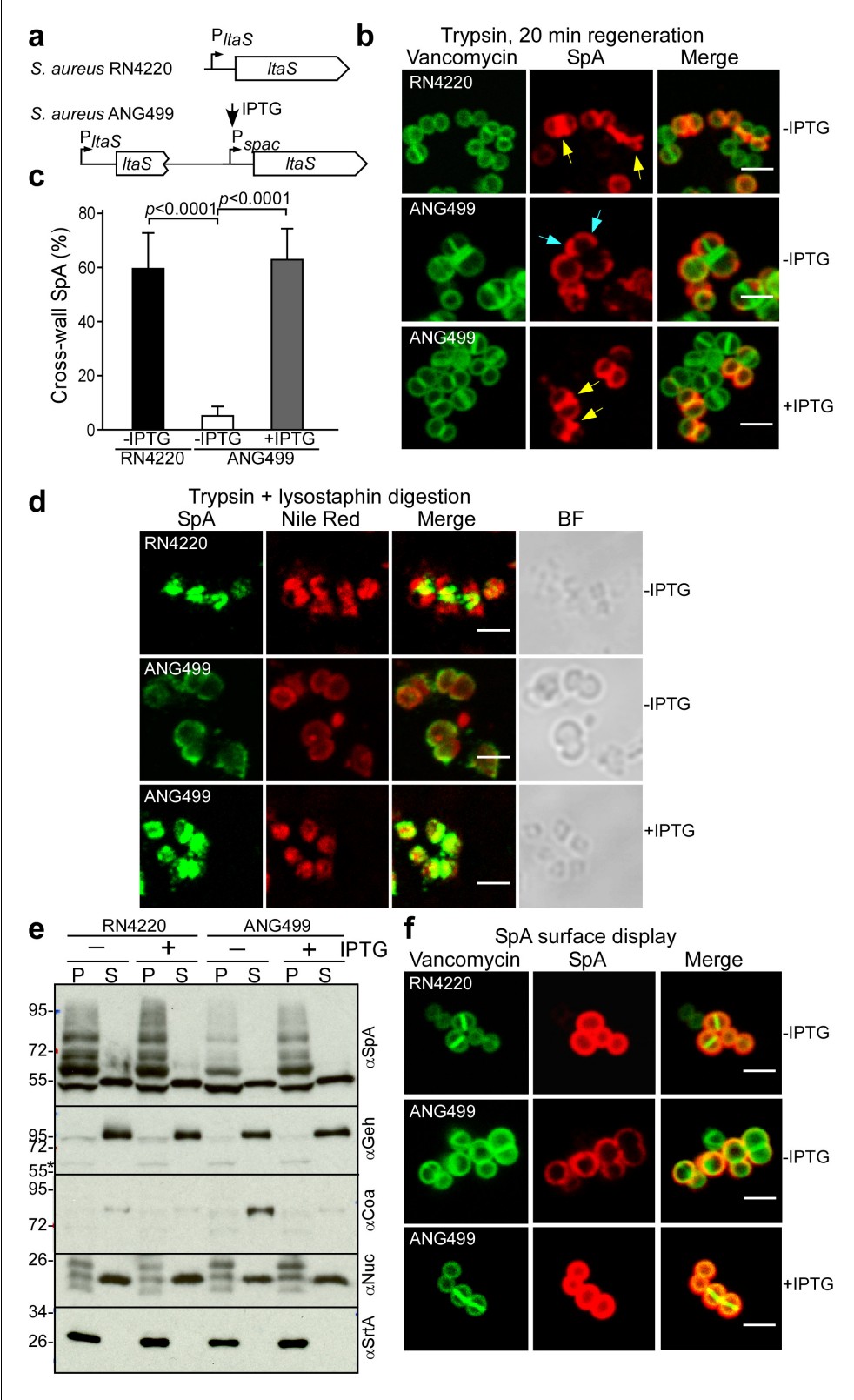

**Figure 10.** Localization of SpA secretion in LtaS-depleted *S. aureus*. (**a**) Schematic to illustrate the *ltaS* locus in *S. aureus* RN4220 and ANG499. (**b**) Fluorescence microscopy with BODIPY-FL-vancomycin (green) and αSpA (red) stained samples 20 min after trypsin removal of surface proteins from the staphylococcal envelope to detect newly synthesized SpA. Scale bar, 2 μm. (**c**) SpA-positive staphylococci in images derived from samples in (**b**) were analyzed for SpA deposition at the cross wall of diplococci (n = 200). Data from three independent experiments were used to derive the mean (± SEM)

*Figure 10 continued on next page*

*Figure 10 continued*
and were analyzed for significant differences with one-way ANOVA for comparisons between *S. aureus* RN4220 (WT) and ANG499 grown with (+LtaS) and without IPTG (-LtaS). *p* values were recorded: RN4220-IPTG vs. ANG499-IPTG, p<0.0001; ANG499-IPTG vs. ANG499 +IPTG, p<0.0001. (**d**) Fluorescence microscopy to localize intracellular SpA in *S. aureus* strains RN4220 (WT) and ANG499 (P$_{spac}$-*ltaS*) grown with and without IPTG induction for 3 hr. Bacteria were trypsin treated to remove extracellular surface proteins and fixed with para-formaldehyde. Samples were then treated with lysostaphin, incubated with detergent and SpA-specific rabbit antibodies and Alexa Fluor 488-labeled goat-anti-rabbit-IgG (green) and with Nile red to reveal bacterial membranes. BF identifies the bright-field microscopy view of fluorescence microscopy images. Scale bar, 2 µm. (**e**) The culture supernatant (S) and bacterial pellet (P) samples of *S. aureus* RN4220 and ANG499 grown for three hours in the presence or absence of IPTG were immunoblotted with antibodies specific for SpA (αSpA), glycerol-ester hydrolase (αGeh), coagulase (αCoa), nuclease (αNuc) and sortase A (αSrtA). (**f**) Fluorescence microscopy of staphylococci to measure surface display of protein A in bacteria stained with BODIPY-FL-vancomycin (green) and αSpA (red) without trypsin treatment. Scale bar, 2 µm.
DOI: https://doi.org/10.7554/eLife.34092.013

hydrophobic core may enable staphylococci to remove the products of processed signal peptides from the membrane. For example, products of degradative proteolysis have been observed during processing of SpA LPXTG motif sorting signal, which is cleaved between Thr (T) and Gly (G) and within the hydrophobic core (*Navarre and Schneewind, 1994*).

Precursors with amino acid substitutions or deletions in the YSIRK/GXXS motif are thought to accumulate in septal membranes, however these variants typically exhibit diminished secretion and cell wall anchoring in the cross wall (*DeDent et al., 2008*; *Yu and Götz, 2012*). Our observations corroborate these findings and suggest that not all features of the YSIRK/GXXS motif are required for precursor targeting to septal membranes. We took advantage of the SpA$_{ED/S18L}$ precursor and used affinity chromatography to purify crosslinked proteins. Several crosslinked proteins were already known to be located in septal membranes (PBP2, EzrA, LtaS), consistent with SpA$_{ED/S18L}$ precursor accumulation in this compartment. Among the crosslinked proteins are two components of the secretion machinery, SecA and SecDF, as well as LtaS, which catalyzes the synthesis of lipoteichoic acid in septal membranes (*Tsirigotaki et al., 2017*; *Percy and Gründling, 2014*). SecA appears to bind directly to the signal peptide of SpA$_{ED/S18L}$. The resulting complex may even engage SecYEG translocase as well as SecDF. However, due to the S18L substitution SpA$_{ED/S18L}$ precursors cannot efficiently enter the septal secretion pathway. It seems unlikely that LtaS may directly interact with SpA$_{ED/S18L}$. We think it is plausible that SpA$_{ED/S18L}$ complexed with SecA and SecYEG at septal membranes can be crosslinked to other septal proteins.

The subcellular localization of the Sec apparatus has been examined in streptococci, enterococci and in *Bacillus subtilis*. A spiral pattern of Sec translocase has been reported for *B. subtilis* (*Campo et al., 2004*). In *S. pyogenes*, contradictory results have been reported for immunogold-labelling and electron microscopy experiments: SecA was localized to a single microdomain and also found distributed throughout the plasma membrane (*Carlsson et al., 2006*; *Rosch and Caparon, 2004*). In *S. pneumoniae*, SecA localization changed during cell cycle progression. In early divisional cells, SecA was predominantly localized to septal membranes, whereas during later stages of division SecA was hemispherically distributed within the region between septa and at the future equators of dividing cells (*Tsui et al., 2011*). *Streptococcus agalactiae* SecA was localized to septal membranes, whereas SecA was detected as a single microdomain in *Streptococcus mutans* and *Enterococcus faecalis* (*Brega et al., 2013*; *Hu et al., 2008*; *Kline et al., 2009*). We show here that *S. aureus* SecA is localized to the plasma membrane and is not spatially restricted to septal membranes or microdomains. This distribution is consistent with our proposed role of SecA, promoting precursor translocation at polar and septal membranes.

When studied for its contribution to septal secretion, SecDF chaperone allows large amounts of protein A to be deposited into the cross-wall peptidoglycan and promotes secretion of YSIRK/GXXS motif precursors (SpA and Geh). Nevertheless, *secDF* is not essential for septal targeting or secretion of SpA precursors. In contrast, cells depleted for SecA, accumulate SpA precursors that cannot traffic to septal membranes in the cytoplasm. Finally, LtaS-depleted staphylococci are unable to synthesize lipoteichoic acid and cannot direct precursors to the septal area. Instead, SpA is directed to polar membranes. We have incorporated these observations into a model whereby septal accumulation of LtaS and of lipoteichoic acids functions as a determinant for SecA-mediated targeting of SpA precursors. Following precursor cleavage at the YSIRK/GXXS motif, truncated SpA (SpA-2) is moved

across the membrane, aided by the proton-motif force and by the chaperone activity of SecDF. Once translocated, SpA is cleaved by signal peptidase to generate SpA-4 and by sortase at the LPXTG motif of its C-terminal sorting signal (*Navarre and Schneewind, 1994*; *Ton-That et al., 1999*). The resulting sortase-acyl intermediate is then incorporated into cross wall peptidoglycan for distribution on the bacterial surface (*Schneewind et al., 1995*).

*Enterococcus faecalis* Eep, a membrane zinc-metalloprotease, cleaves signal peptides of lipoprotein precursors into short hydrophobic peptides that are subsequently dislodged from the membrane by an ABC transporter and function as pheromones to induce the transfer of conjugative plasmids by specific donors (*An et al., 1999*; *Chandler and Dunny, 2008*; *Varahan et al., 2014*). The corresponding zinc-metalloprotease in *S. aureus* is designated RseP (SAOUHSC_01239). However, RseP is not involved in septal secretion and cleavage of the YSIRK-GXXS signal peptides, as a deletion of the corresponding gene does not affect SpA septal secretion in the *S. aureusΔSAOUHSC_01239* mutant (*Figure 4—figure supplement 1*).

## Materials and methods

### Bacterial strains and growth conditions

*E. coli* strains were grown in Luria-Bertani broth (LB) or LB agar. *S. aureus* strains were grown in tryptic soy broth (TSB) or agar (TSA). Ampicillin (100 µg/ml) was used for plasmid selection in *E. coli*. Chloramphenicol was used for selection of pOS1 derivatives (10 µg/ml) and pCL55 derivatives (5 µg/ml) in *S. aureus* (*Lee et al., 1991*). Erythromycin (Erm 10 µg/ml) was used for selection of *ermB* marked *bursa aurealis* transposon mutants in *S. aureus* WY110 (*Δspa Δsbi*) and 10 µg/ml Erm plus 1 mM isopropyl β-D-1-thiogalactopyranoside (IPTG) was used for pMutin–HA-5'secA selection in *S. aureus*. Expression from the $P_{spac}$ promoter was induced with 1 mM IPTG. Anydrotetracycline (ATc, 200 ng/ml) was used to induce expression from the tetracycline-inducible promoter in pCL55-$P_{tet}$ constructs.

### Plasmids and strains

To avoid mutations in the *spa* gene, all cloning procedures were performed at 30°C. All pOS1-derivative and pCL55-derivative plasmids were constructed in *E. coli* DC10B (*Monk et al., 2012*) and transformed to *S. aureus* strains by electroporation (*Schneewind and Missiakas, 2014*). All plasmids and strain constructs were validated by DNA sequencing (*Supplementary file 2*). Primers used in this study are listed in *Supplementary file 3*. To avoid cross reaction in SpA immunoblot and purification, *S. aureus* WY110 (*Δspa Δsbi*) was generated by transducing *the sbi::ermB* allele from the Phoenix library (*Bae et al., 2004*) into *S. aureus* SEJ1, that is *S. aureus* RN4220 carrying *Δspa*. Phage transduction was performed as described previously (*Schneewind and Missiakas, 2014*). To construct pSpA$_{ED}$, primers 10 and 69 were used to amplify the *spa* promoter and *spa$_{ED}$* coding sequence encoding IgBDs E and D (−40 bp upstream of the transcription start site of *spa* to 459 bp of *spa* coding sequence) from chromosomal DNA of *S. aureus* RN4220. The PCR product was digested with EcoRI and BamHI, and ligated with plasmid pOS1 (*Schneewind et al., 1993*). To generate mutations and deletions within the SpA signal peptide sequence, quick-change mutagenesis was performed as follows: primers pairs (*Supplementary file 3*) that contain desired mutation or deletion were used to PCR amplify pSpA$_{ED}$. The PCR products were digested with DpnI and transformed to *E. coli* DC10B. Plasmid variants confirmed by DNA sequencing were transformed into *S. aureus* WY110. To construct pCL55-SpA and its derivatives, primer pairs 175 and 177 were used to PCR amplify the *spa* promoter and full-length *spa* coding sequence. PCR products were digested with BamHI and KpnI and ligated into pCL55 cut with the same enzymes (*Lee et al., 1991*). The resulting plasmid, pCL55-SpA, was used as template for PCR mutagenesis of its signal peptide mutant derivatives via quick-change mutagenesis as described above. To construct pCL55-SpA$_{SP-SasF}$, primer pairs 175 and 21 were used to amplify the promoter sequence of *spa*. Primers 22 and 23 were used to amplify coding sequence for the signal peptide sequence of *sasF*. Last, primers 24 and 177 were used amplify *spa* coding sequence for the E and D IgBDs. All three DNA fragments were ligated via SOE (splicing by overlap extension) PCR, digested with BamHI and KpnI, and then ligated into pCL55 cut with the same restriction enzymes. pCL55-derivaties were transformed into *S. aureus* WY110 and integrated into the chromosome at the *geh* locus (*Lee et al., 1991*). The integration was

confirmed by PCR. To construct the *secA* depletion strain *S. aureus* WY223 (P$_{spac}$-*secA*), primers 189 and 190 were used to amplify the ribosome binding site and the first 656 bp of the *secA* gene. The PCR product was digested with HindIII and KpnI and ligated with pMutin–HA (Bacillus Genetic Stock Center, Columbus, OH). The resulting plasmid pMutin–HA-5'secA was transformed into RN4220 and integrated at the *secA* locus in the chromosome. Clones were selected on TSA supplemented with 10 µg/ml erythromycin and 1 mM IPTG. To construct pCL55-P$_{tet}$-*secA:sfGFP*, primers 180 and 181 were used to amplify *secA* full-length coding sequence together with its ribosome binding site. Primers 182 and 183 were used to amplify *sfGFP* gene from pCX-sfGFP (*Yu and Götz, 2012*). The two DNA fragments were joined together by SOE. The resulting *secA:sfGFP* hybrid, which contains the 'Gly-Gly-Ala-Ala-Gly-Ala' between SecA and sfGFP, was digested with AvrII-BglII and ligated with pCL55-P$_{tet}$ (*Gründling and Schneewind, 2007*). pCL55-P$_{tet}$-*secA:sfGFP* was transformed to into *S. aureus* WY223 (P$_{spac}$-*secA*) and integrated into the chromosome at the *geh* locus, thereby generating *S. aureus* WY230 (P$_{spac}$-*secA*, P$_{tet}$-*secA:sfGFP*). Plasmid pKOR1 based allelic replacement (*Bae and Schneewind, 2006*) was used to generate the Δ*secDF* (*S. aureus* WY418) and Δ*rnd2* (*S. aureus* WY416) knock-out mutants. *S. aureus* WY412, a mutant with Δ*secDF* Δ*rnd23* mutations, was generated by transducing the *rnd3::ermB* allele from the Phoenix library into *S. aureus* carrying Δ*secDF* and Δ*rnd2* mutations. To construct the complementation plasmid pSecDF, the *secDF* ORF and 274 bp upstream sequence were PCR amplified with primers 315 and 316, digested with EcoRI and BamHI, and ligated into pOS1 cut with the same enzymes. The resulting plasmid, pSecDF, was transformed into *S. aureus* strains WY418 (Δ*secDF*) and WY412 (Δ*secDF* Δ*rnd23*).

## Cell fractionation and immunoblotting

Bacterial overnight cultures were diluted 1: 100 into fresh TSB and grown to OD$_{600}$ 0.8. One ml culture was centrifuged at 18,000 × *g* for 5 min in an Eppendorf tube. The culture supernatant (S) was transferred to another tube and proteins were precipitated with 10% trichloroacetic acid (TCA) on ice for 30 min. The bacterial sediment (P, pellet) was suspended in 1 ml Tris-buffer [50 mM Tris-HCl (pH 7.5), 150 mM NaCl] and incubated with 20 µg/ml lysostaphin at 37°C for 30 min. After cell lysis, proteins from the cell pellet were precipitated with 10% TCA. To localize proteins in different cellular compartments, cell fractionation was performed as follows: 1 ml culture (OD$_{600}$ = 0.8) was centrifuged at 18,000 × *g* for 5 min in an Eppendorf tube. The supernatant was transferred to another tube and proteins were precipitated with 10% TCA (S, supernatant). The pellet was suspended in 1 ml TSM [50 mM Tris-HCl (pH 7.5), 0.5 M sucrose, 10 mM MgCl$_2$] and incubated with 20 µg/ml lysostaphin for 10 min at 37°C. After centrifugation at 18,000 × *g* for 5 min, the supernatant (cell wall fraction) was transferred to another tube. The protoplast pellet was suspended in 1 ml Tris-buffer [50 mM Tris-HCl (pH 7.5), 10 mM MgCl$_2$] and subjected to three freeze-thaw cycles in dry ice/ethanol and warm water baths. Membranes in cell lysates were sedimented by ultracentrifugation 150,000 × *g* for 40 min. Supernatant was transferred to another tube (cytosolic fraction), whereas the pellet (membrane fraction) was suspended in 1 ml Tris-buffer and precipitated with 10% TCA. After TCA precipitation on ice for 30 min, proteins were sedimented at 18,000 × *g* for 10 min, washed with ice-cold acetone, air-dried and solubilized in 100 µl 1 × SDS sample buffer [62.5 mM Tris-HCl (pH 6.8), 2% SDS, 10% glycerol, 5% 2-mercaptoethanol, 0.01% bromophenol blue]. For immunoblotting, protein samples were separated on 10, 12 or 15% SDS-PAGE and transferred to polyvinylidene difluoride (PVDF) membranes. Membranes were blocked with 5% milk for 45 min. As needed, 50 µl human IgG (Sigma) was added to 10 ml block-solution to block SpA cross-reaction. Primary antibodies were affinity-purified rabbit polyclonal antibodies against SpA$_{KKAA}$ (1:10,000 dilution), rabbit serum of anti-SrtA (1:20,000 dilution), rabbit serum of anti-SecA (1:10,000 dilution), rabbit serum of anti-Geh (1:10,000 dilution), polyclonal antibodies of anti-Coa (1:5000 dilution), rabbit serum of anti-Nuc (1:5000 dilution), and anti-GFP rabbit serum (1:10,000 dilution) (Invitrogen). Membranes were incubated with primary antibodies for 1 hr, washed three times for 5 min with TBST [50 mM Tris-HCl (pH 7.5), 150 mM NaCl, 0.1% Tween 20], incubated with secondary anti-rabbit IgG linked to horseradish peroxidase (HRP) for 1 hr, washed, and developed using enhanced chemiluminescence substrates. The intensity of immunoblot signals was analyzed and measured with Image J software (*Schneider et al., 2012*). Statistical analysis was performed using GraphPad Prism software. One-way ANOVA (Dunnett's multiple comparisons test) was used to compare the mean for each variant with the mean for SpA$_{ED}$ wild-type (*Figures 1c* and *2b*).

## Pulse-labeling

Staphylococcal cultures were grown to mid-log phase ($OD_{600}$ 0.8) in TSB and bacteria sedimented by centrifugation at 18,000 × $g$ for 5 min. Bacterial pellets were washed twice and suspended in 1 ml minimal medium 4. [$^{35}$S]methionine/cysteine (100 µl = 100 µCi Perkin Elmer) was added to bacterial suspensions, vortexed and incubated for 60 s at 37°C. 250 µl was removed and immediately mixed with 250 µl ice-cold 10% TCA to quench all metabolic activity (time 0'). Chase solution (50 µl of 2 mg/ml methionine, 2 mg/ml cysteine and 10 mg/ml casamino acids) was added to the remainder of bacterial suspension and incubated for 1, 5 and 20 min. At each time point, 250 µl bacterial suspension was removed and mixed with 250 µl ice-cold 10% TCA. TCA precipitated cells were washed with acetone, dried and suspended in 1 ml 0.5 M Tris-HCl (pH 7.0) containing 20 µg/ml lysostaphin. After lysostaphin treatment at 37°C for 1 hr, cell lysate was precipitated with 7% TCA, washed with acetone, dried, suspended in 50 µl 4% SDS, 0.5 M Tris-HCl (pH 7.5) and allowed to incubate for 30 min prior to boiling. Subsequently, samples were incubated for 1 hr with rabbit polyclonal anti-SpA$_{KKAA}$ antibody (*Kim et al., 2010*) that was 1:1000 diluted in 1 ml RIPA buffer (0.1% SDS, 0.5% deoxycholic acid, 1% Triton X-100, 50 mM Tris-HCl pH 8.0, 150 mM NaCl). Protein A sepharose (50 µl of 50% slurry, Sigma) was added to each sample and incubated for 1 hr followed by five washes with 1 ml RIPA buffer. Proteins bound to the beads were solubilized by boiling in 15 µl 2 × SDS sample buffer for 10 min and separated on 10% (SpA) or 15% SDS-PAGE (SpA$_{ED}$). Gels were dried on Whatman 3 M paper and autoradiographed on X-ray film for 48 hr or longer.

## Purification of SpA$_{ED/S18L}$, Edman degradation and MALDI-TOF mass spectrometry

Overnight bacterial cultures of *S. aureus* WY110 (pSpA$_{ED}$ or its derivatives) were diluted 1:100 into 4 liters of TSB and grown to $OD_{600}$ 2. Staphylococci were sedimented by centrifugation at 8,000 × $g$ for 10 min. Bacteria were suspended in 30 ml of Tris-buffer, 0.5 (vol/vol) 0.1 mm sterilized glass beads were added and peptidoglycan was broken with 15 × 1 min pulses in a bead-beating instrument (MP Biomedicals). Samples were centrifuged at 7,000 × $g$ for 10 min to sediment glass beads. The supernatant was transferred to another tube and centrifuged at 33,000 × $g$ for 1 hr at 4°C. The membrane sediment was suspended in 30 ml RIPA buffer and incubated for 1 hr with rotation. RIPA extract was centrifuged at 33,000 × $g$ for 1 hr at 4°C. The supernatant was removed and subjected to affinity chromatography. Two ml 50% suspension of IgG sepharose (GE Healthcare) was loaded onto each column. The column bed was washed once with 7 ml 0.1 M glycine (pH 3.0), twice with 14 ml 50 mM Tris-HCl (pH 7. 5) and once with 10 ml RIPA buffer. RIPA membrane extracts were loaded onto the column followed by two washes with 14 ml RIPA buffer and once with 10 ml 50 mM Tris-HCl (pH 7. 5). Proteins were eluted by adding four times 1 ml 0.1 M glycine (pH 3.0) to the column and immediately neutralizing the eluate with 25 µl of 1.5 M Tris (pH 8.8). For Edman degradation, the purified SpA$_{ED}$ precursors were 10-fold concentrated via Amicon Ultra-0.5 ml Centrifugal Filters (10 kD cut off). Proteins were separated on 15% SDS-PAGE, electro-transferred to PVDF and stained with Coomassie-Brilliant Blue. Bands of interest were excised and subjected to Edman degradation (Alphalyse, Inc, CA, USA). For MALDI-TOF mass spectrometry analysis, 1 µl of SpA$_{ED}$ sample was mixed with 1 µl of 10 mg/ml sinapic acid, dried on the Bruker MTP 384 plate, and examined in a Bruker Autoflex Speed MALDI-TOF mass spectrometer in the linear positive-ion mode using peptide standards for calibration.

## Crosslinking of SpA$_{ED}$ precursor

Overnight cultures of *S. aureus* WY110 (Δ*spa* Δ*sbi*, pSpA$_{ED/S18L}$) and *S. aureus* WY110 (Δ*spa* Δ*sbi*, pSpA$_{ED/SP-SasF}$) were each diluted 1: 100 into 4 L TSB and grown to $OD_{600}$ 2. Formaldehyde (0.9%, methanol free) was added to the bacterial culture and incubated for 20 min with shaking. Cross-linking was quenched by adding 400 ml ice-cold 0.125 M glycine and rotating the sample for 10 min. Staphylococci were sedimented by centrifugation at 8,000 × $g$ for 10 min. Bacteria were suspended in 30 ml of 50 mM Tris-HCl (pH 7.5), 150 mM NaCl, and washed twice in the same buffer. Sterilized 0.1 mm glass beads 0.5 (vol/vol) were added and peptidoglycan broken with 15 × 1 min pulses in a bead-beating instrument (MP Biomedicals). Samples were centrifuged at 7000 × $g$ for 10 min to sediment glass beads. The supernatant was transferred to another tube and centrifuged at 33,000 × $g$ for 1 hr at 4°C. The membrane sediment was suspended in 30 ml 50 mM Tris-HCl (pH 7. 5), 2%

n-dodecyl β-D-maltoside (DDM) and incubated at 4°C overnight. Samples were subjected to ultra-centrifugation at 150,000 × g for 40 min. The supernatant was subjected to affinity chromatography on IgG sepharose affinity purification as described above. Eluate was concentrated via Amicon Ultra-0.5 ml 10 kD Centrifugal Filters and mixed with equal volume of 2 × SDS sample buffer. To reverse the cross-linking, samples were either boiled at 95°C for 20 min or, as a control, incubated at 60°C (no reversal of cross-linking). Proteins in all samples were separated on 12% SDS-PAGE and bands of interest excised as indicated in *Figure 4* and subjected to protein identification and semi-quantitative analysis at the Harvard University Taplin Mass Spectrometry Facility (*Supplementary file 1*).

## SecA depletion and SecA-sfGFP induction

Overnight cultures of *S. aureus* RN4220 (WT), *S. aureus* WY223 (P*spac*-*secA*) and *S. aureus* WY230 (P*spac*-*secA*, P*tet*-*secA:sfGFP*) were grown in TSB and 1 mM IPTG. Overnight cultures were washed twice with an equal volume of TSB and diluted 1:100 into fresh TSB with or without 1 mM IPTG and with or without 200 ng/ml ATc. After 3 hr growth at 37°C, cultures were diluted into fresh TSB with or without IPTG or ATc and subjected to further growth at 37°C. Growth was monitored by sampling cultures at timed intervals and measuring optical density. One ml bacterial culture was removed after 3 hr (prior to the second 1:100 dilution) and after 6 hr (3 hr after the second 1:100 dilution). Samples were processed for protein secretion and immunoblotting assays or analyzed by fluorescence microscopy. A similar procedure was performed for LtaS depletion using *S. aureus* ANG499 (P*spac*-*ltaS*). Samples from the LtaS depletion experiments were analyzed after 3 hr growth with or without 1 mM IPTG.

## Fluorescence microscopy

To observe SpA targeting to the cross-wall, 2 ml mid-log phase *S. aureus* culture (OD$_{600}$ 0.8) were centrifuged at 18,000 × g for 5 min, supernatant removed, bacteria washed once in 2 ml PBS and suspended in 1 ml PBS containing 0.5 mg/ml trypsin (Sigma). After incubation at 37°C for 1 hr, staphylococci were washed twice with PBS, suspended in fresh TSB containing 2.5 mg/ml soybean trypsin inhibitor (Sigma) and incubated at 37°C for 20 min with rotation. 250 µl of the cell suspension was removed and immediately mixed with fixation solution (2.5% paraformaldehyde and 0.006% glutaraldehyde in PBS). The cells were fixed for 20 min at room temperature, washed three times with PBS and applied to poly-L-lysine coated 8-well glass slides (MP Biomedicals) for 5 min. Excess and non-adherent cells were washed away with PBS. Immobilized cells were blocked with 3% BSA in PBS for 45 min and incubated with SpA-specific mouse hybridoma monoclonal antibody 5A10 (*Kim et al., 2010*) (diluted 1:4000 in 3% BSA) for 1 hr. Cells were washed eight times with PBS and further incubated in the dark with Alexa Fluor 647 conjugated anti-mouse IgG (1:500 in 3% BSA) (Invitrogen, Carlsbad, CA) for 1 hr. Cells were washed 10 times with PBS and incubated with 1 µg/ml BODIPY-FL vancomycin (ThermoFisher) for 10 min in the dark followed by washing five times with PBS. A drop of SlowFade Gold reagent (Molecular Probes) was applied to samples before sealing coverslips with nail polish. Fluorescent images were visualized and captured on a Leica SP5 Tandem Scanner Spectral 2-Photon Confocal microscope with 100 × oil objective. Identical settings and exposure times were applied to all samples.

To image SpA display on the staphylococcal surface, 1 ml of mid-log phase *S. aureus* cultures were centrifuged at 18,000 × g for 5 min and supernatant removed. Bacteria were washed once in 2 ml PBS and suspended in 1 ml PBS and mixed with fixation solution. Cells were fixed for 20 min at room temperature, washed three times with PBS and applied to poly-L-lysine coated 8-well glass slides (MP Biomedicals) for 5 min, stained with vancomycin and αSpA antibodies and analyzed by fluorescence microscopy.

To localize intracellular SpA, 2 ml of mid-log phase staphylococcal cultures were centrifuged at 18,000 × g for 5 min and supernatant removed. Bacteria were washed once in 2 ml PBS and suspended in 1 ml PBS, 0.5 mg/ml trypsin (Sigma). After incubation at 37°C for 1 hr, staphylococcal cells were washed twice with PBS and fixed with fixation solution. The cells were fixed for 15 min at room temperature and 30 min on ice, washed three times with PBS and suspended in 1 ml GTE buffer [50 mM glucose, 20 mM Tris-HCl (pH 7.5), 10 mM EDTA]. Lysostaphin (10 µg/ml) was added and 50 µl cell suspensions were immediately applied to poly-L-lysine coated 8-well glass slides and incubated for 1 min. Non-adherent cells were removed and PBS, 0.2% Triton X-100 was applied to samples for

10 s. Excessive liquid was aspirated and slides were air-dried. Dried slides were immediately dipped in methanol at −20°C for 5 min, and in acetone at −20°C for 30 s and then allowed to dry completely. Afterwards, the cells on the slides were re-hydrated with PBS for 5 min, blocked with 3% BSA, stained with the membrane dye Nile red (Sigma) and rabbit antibodies specific for SpA followed by Alexa-Fluor-488 conjugated goat-anti-rabbit-IgG and analyzed by fluorescence microscopy as described above.

To visualize the sub-cellular localization of SecA-sfGFP, samples from 3 and 6 hr growth cultures were removed as described above. Bacteria were sedimented by centrifugation and washed twice in PBS. Cells were stained with 1 μg/μl FM4-64FX (Molecular Probes) for 10 min in the dark and were then fixed with fixation solution for 20 min. After washing twice with PBS, cells were applied to poly-L-lysine coated coverslips, and incubated for 5 min. After removing excess bacterial cells, SlowFade Gold reagent was added to the samples and the coverslips were sealed onto glass slides. Samples were visualized under Leica SP8 3D, 3-color Stimulated Emission Depletion (STED) laser scanning confocal microscope with 100×/1.45 oil objective. Images were captured with identical settings. Deconvolution images with identical parameters were generated by using a HyVolution module installed on the microscope.

All the images were analyzed in Image J software (*Schneider et al., 2012*). To quantify the frequency of SpA cross-wall localization at 20 min regeneration after trypsin digestion, numbers of diplococci and numbers of cross-wall localized SpA were counted manually using the cell counter tool in Image J. Diplococci were defined as two daughter cells that had divided and formed a cross-wall but had not yet separated. Cross-wall localized SpA signals were defined as lines at the cross-wall of diplococci. Diplococci were counted in vancomycin stained images and cross-wall localized SpA was counted in merged images. The frequency was determined by dividing cross-wall localized SpA by the number of diplococci. An example of the counting method is displayed in the Source file to *Figure 2e*. At least two random images were acquired per sample for each experiment. Three or more independent experiments were performed and data from more than 200 diplococci were analyzed for statistically significant differences using one-way ANOVA with Dunnett's multiple comparison test comparing staphylococci expressing wild-type and *spa* variants (*Figure 2e*). In *Figure 6c* and *Figure 10c* the Tukey's multiple comparison test was used to analyze differences between multiple groups.

## Acknowledgements

We thank Vytas Bindokas (Microscopy Core Facility, University of Chicago) for assistance with microscopy and members of our laboratory for experimental advice and discussion. This work was supported by National Institute of Allergy and Infectious Diseases grants AI038897 and AI052474. WY acknowledges support from German Research Foundation (DFG) Fellowship (award YU 181/1–1).

## Additional information

### Funding

| Funder | Grant reference number | Author |
| --- | --- | --- |
| National Institute of Allergy and Infectious Diseases | AI038897 | Olaf Schneewind |
| Deutsche Forschungsgemeinschaft | YU 181/1-1 | Wenqi Yu |
| National Institute of Allergy and Infectious Diseases | AI052474 | Olaf Schneewind |

The funders had no role in study design, data collection and interpretation, or the decision to submit the work for publication.

## Author contributions

Wenqi Yu, Conceptualization, Data curation, Formal analysis, Funding acquisition, Investigation, Methodology, Writing—original draft, Writing—review and editing; Dominique Missiakas, Conceptualization, Data curation, Formal analysis, Supervision, Investigation, Visualization, Methodology, Writing—original draft, Project administration, Writing—review and editing; Olaf Schneewind, Conceptualization, Resources, Data curation, Formal analysis, Supervision, Funding acquisition, Validation, Investigation, Methodology, Writing—original draft, Project administration, Writing—review and editing

## Author ORCIDs

Olaf Schneewind (iD) http://orcid.org/0000-0001-9652-3823

## Decision letter and Author response

Decision letter https://doi.org/10.7554/eLife.34092.019
Author response https://doi.org/10.7554/eLife.34092.020

# Additional files

## Supplementary files

• Supplementary file 1. List of ESI-MS identified tryptic peptides crosslinked to SpA$_{ED/S18L}$.
DOI: https://doi.org/10.7554/eLife.34092.014

• Supplementary file 2. Strains and plasmids used in this study.
DOI: https://doi.org/10.7554/eLife.34092.015

• Supplementary file 3. Oligonucleotide primers used in this study.
DOI: https://doi.org/10.7554/eLife.34092.016

• Transparent reporting form
DOI: https://doi.org/10.7554/eLife.34092.017

## Data availability

All data generated or analysed during this study are included in the manuscript and supporting files. Source data files have been provided for Figure 2.

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
