## [Decision Letter]

Thank you for submitting your article "Septal Secretion of Protein A in *Staphylococcus aureus* Requires SecA and Lipoteichoic Acid Synthesis" for consideration by *eLife*. Your article has been reviewed by two peer reviewers, and the evaluation has been overseen by a Reviewing Editor and Gisela Storz as the Senior Editor. The following individual involved in review of your submission has agreed to reveal his identity: Richard Novick (Reviewer #2).

The reviewers have discussed the reviews with one another and the Reviewing Editor has drafted this decision to help you prepare a revised submission.

This is very thorough and detailed analysis of the role of the YSIRK/GXXS motif in septal secretion in staphylococci and, by implication, in other /gram-positive bacteria. It will add substantively to our understanding of protein secretion in the organisms.

Two reviewers carefully read and commented on the manuscript. Revisions should be straightforward. Reviewer 3 has three major concerns, which I anticipate you will be able to address. The first of these concerns is a sort of organizational issue. Dr. Novick has posed two questions, which you might want to address in your revised manuscript. Normally, we would consolidate the reviews into one document indicating essential revisions, but everything seems to be very straightforward here and we are just including the two reviews in their entirety.

*Reviewer #2:*

This is very thorough and detailed analysis of the role of the YSIRK/GXXS motif in septal secretion in staphylococci and, by implication, in other /gram-positive bacteria. It will add substantively to our understanding of protein secretion in the organisms.

I have only a few suggestions for improvement of the paper, plus 2 major questions, of which the answers would add nicely to the presentation (but are not really required): why is SecA required for growth, and which proteases catalyze the cleavages within the signal peptide?

Abstract and the last paragraph of the Introduction – confusing and difficult to understand; does not really convey a clear message – should be rewritten.

Introduction, first paragraph – are all proteins with the YSIRK/GXXS motif trafficked to the septal membrane?

Introduction, second paragraph – "relieved"? Replace with "cleaved" or "disrupted".

Subsection “SpA signal peptide variants defective for septal secretion”, first paragraph – is the nature of this faster-moving precursor known?

Subsection “SpA signal peptide variants defective for septal secretion”, last paragraph – delete "encoding the" and "variants".

Subsection “SpA signal peptide variants defective for septal secretion”, last paragraph – explain "SpASP-SasF".

Subsection “Processing of SpA signal peptide variants” – the hydrophobic portion of staphylococcal signal peptides have been observed to serve as sex pheromones promoting conjugation with enterococci. Could this be the role of the peptide produced by this cleavage? What protease is responsible for the YSIRK/GXXS cleavage?

Subsection “SecA depletion in *S. aureus*” – Figure 5B does not appear to show the patterns described in the text – e.g. there is no growth arrest of secA-depleted cells, and it is noted later that there is residual SecA is present. These points should be made consistent.

Subsection “SecA depletion blocks SpA secretion”, last paragraph – any idea why secA mutants are defective for growth? Could it be required for cell division as perhaps suggested by the observations described in the last paragraph of the subsection “Localization of SecA and SpA precursors in staphylococci”?.

*Reviewer #3:*

The gap in knowledge that this study aims to address is: How are YSIRK/GXXS-containing proteins directed to the septal membrane for localized secretion? The authors identify YSIRK/GXXS signal peptide interacting partners and create a beautiful series of conditional mutant strains in order to query the role of a subset of those interacting partners in SpA septal targeting. Overall, the experimental design and execution was very nice and I especially appreciated the abundance of controls at every stage.

The major findings in this study are 1) SpA signal peptide variant interacts with SecA and LtaS, and 2) SecA or LtaS depletion results in SpA not at the septum but diffuse around the cell in the case of SecA – and polar in the case of LtaS (Figure 8 and 10).

Secondary findings in this study are 1) the identification of SpA signal sequence residues that are important for precursor processing and SpA secretion, 2) cleavage sites within the SpA signal sequence, and 3) the unsurprising report that depletion of SecA or deletion of SecDF result in reduced secretion of several secreted proteins.

Major comments:

While the cumulative data in the paper support the title that SpA septal secretion requires SecA, there are numerous instances earlier in the paper, before the key data to support this are shown (in Figure 8) where statements to this effect are made. For example, the first section title indicates SpA signal peptide variants are defective for septal secretion when the data presented in this section only support a general secretion phenotype. There are other premature examples of this and I found it confusing and later misleading. I suggest this be modified throughout the manuscript.

The crosslinking and pulldown experiments used a mutant SpA precursor that appears to accumulate at higher levels than WT in general, and accumulates at the septum. While (I think) I understand the rationale for using this construct for pulldowns (they are enriched at the septum so interactions are more likely), I wonder if it could lead to 'forced' interactions and/or miss other interactions with wild type SpAED. Did the authors also try using SpAED? What happened? Can these interactions be validated in the absence of crosslinks (either for the mutant or WT signal sequence) by bacterial two hybrid or another method, to confirm if they are direct or indirect interactions? And why were SecA, SecDF, and LtaS singled out for further analysis?

I am puzzled by the data presented in Figure 7 and 8: on one hand, SecA is everywhere (Figure 7), but upon depletion (albeit the conditions used here are unclear) SpA becomes diffuse (presumably in the membrane?) (Figure 8). Since the cells don't look sick (as in Figure 6D), this must be a mild SecA depletion. So the residual SecA is presumably everywhere in the membrane, but is not interacting with SpA at the septum? Why not? What provides the specificity for this localized interaction? Why does YSIRK/GXXS interact with only a subset of SecA molecules at the septum when SecA is everywhere? This key question remains unclear to me.

---

## [Author Response]

Reviewer #2:[…] I have only a few suggestions for improvement of the paper, plus 2 major questions, of which the answers would add nicely to the presentation (but are not really required): why is SecA required for growth?

In *E. coli, secA* is required for bacterial growth because the sum of all secreted proteins contributes to bacterial replication. About one third of all proteins are secreted across the plasma membrane or inserted into the plasma membrane. We presume the same applies to *S. aureus*. For example, when *secA* is depleted for 3 hours, staphylococcal growth is not yet impaired albeit that SecA is undetectable by immunoblotting (Figure 5D and Figure 7C). After 6 hours of incubation without production of new SecA, staphylococci cease growth presumably because insufficient amounts of secreted proteins are available for the uptake of nutrients, energy production, membrane hemostasis and cell division. The text has been modified to make this clear (subsection “SecA depletion in *S. aureus*”).

And which proteases catalyze the cleavages within the signal peptide?

We don’t know the answer to this question.

Abstract and the last paragraph of the Introduction – confusing and difficult to understand; does not really convey a clear message – should be rewritten.

We have rewritten this section as requested (Abstract and Introduction, last paragraph).

Introduction, first paragraph – are all proteins with the YSIRK/GXXS motif trafficked to the septal membrane?

Yes. Our previous work demonstrated that all proteins with YSIRK/GXXS signal peptide travel to the staphylococcal septum.

Introduction, second paragraph – "relieved"? Replace with "cleaved" or "disrupted".

We replaced “relieved” with “resolved”.

Subsection “SpA signal peptide variants defective for septal secretion”, first paragraph – is the nature of this faster-moving precursor known?

Yes. We have purified the precursors and identified their amino acid sequence, which correspond to SpA_ED/R10A_-2 and SpA_ED/S18L_-2 in Figure 3.

Subsection “SpA signal peptide variants defective for septal secretion”, last paragraph – delete "encoding the" and "variants".

The text has been changed. We did not use the reviewer’s recommendation, as amino acid substitutions do not represent mutations.

Subsection “SpA signal peptide variants defective for septal secretion”, last paragraph – explain "SpASP-SasF".

SpA_SP_-SasF is now explained in the text.

Subsection “Processing of SpA signal peptide variants” – the hydrophobic portion of staphylococcal signal peptides have been observed to serve as sex pheromones promoting conjugation with enterococci. Could this be the role of the peptide produced by this cleavage? What protease is responsible for the YSIRK/GXXS cleavage?

The reviewer refers to beautiful work in *Enterococcus faecalis*, where Eep, a membrane zinc-metalloprotease, cleaves signal peptides of lipoprotein precursors into short hydrophobic peptides that are subsequently dislodged from the membrane by an ABC transporter and function as pheromones to induce the transfer of conjugative plasmids by specific donors. The corresponding zinc-metalloprotease in *S. aureus* is designated RseP. We already know that RseP is not involved in the cleavage of the YSIRK-GXXS signal peptides (Figure 4—figure supplement 1). We believe this makes sense, as the signal peptides of lipoproteins do not share sequence similarity with the YSIRK-GXXS signal peptide. Earlier work demonstrated that *S. aureus* produces pheromone peptides for mating with *E. faecalis*, however the corresponding peptides are not derived from precursors with YSIRK-GXXS signal peptides. Thus, RseP/Eep is not involved in cleaving YSIRK-GXXS signal peptides and the cleavage products of YSIRK-GXXS signal peptides are not thought to function as pheromone signals for mating. We have introduced some of these arguments in the Discussion (last paragraph) to explain the seemingly parallel events between the two signal peptide cleavage systems. See also data in the new Figure 4**—**figure supplement 1.

Subsection “SecA depletion in S. aureus” – Figure 5B does not appear to show the patterns described in the text – e.g. there is no growth arrest of secA-depleted cells, and it is noted later that there is residual SecA is present. These points should be made consistent.

We have expanded the text to say that growth arrest in SecA-depleted staphylococci occurs after 6 hours of incubation, not after 3 hours, and that SecA product is depleted as early as 3 hours.

Subsection “SecA depletion blocks SpA secretion”, last paragraph – any idea why secA mutants are defective for growth? Could it be required for cell division as perhaps suggested by the observations described in the last paragraph of the subsection “Localization of SecA and SpA precursors in staphylococci”?.

Please read our first response above.

Reviewer #3:[…] While the cumulative data in the paper support the title that SpA septal secretion requires SecA, there are numerous instances earlier in the paper, before the key data to support this are shown (in Figure 8) where statements to this effect are made. For example, the first section title indicates SpA signal peptide variants are defective for septal secretion when the data presented in this section only support a general secretion phenotype. There are other premature examples of this and I found it confusing and later misleading. I suggest this be modified throughout the manuscript.

We appreciate the reviewers concerns and have modified the manuscript accordingly.

The crosslinking and pulldown experiments used a mutant SpA precursor that appears to accumulate at higher levels than WT in general, and accumulates at the septum. While (I think) I understand the rationale for using this construct for pulldowns (they are enriched at the septum so interactions are more likely), I wonder if it could lead to 'forced' interactions and/or miss other interactions with wild type SpAED. Did the authors also try using SpAED? What happened? Can these interactions be validated in the absence of crosslinks (either for the mutant or WT signal sequence) by bacterial two hybrid or another method, to confirm if they are direct or indirect interactions? And why were SecA, SecDF, and LtaS singled out for further analysis?

The reviewer raised two excellent points. First, is it possible to capture physiological interactions between a substrate (here the secretion substrate, i.e. YSIRK-GXXS signal peptide bearing precursor) and a catalyst (here the translocation machinery, minimally composed of SecA and SecYEG)? We presume most enzymologists would answer such query with “depends on the rate of catalysis”. Translocation, similar to translation, is thought to be fast: it takes seconds to translocate a precursor several hundred residues in length. One might therefore presume that the answer to such query is “no”. This did not prevent us from trying. Using affinity chromatography of wild-type SpA_ED_ solubilized from membranes to enrich for precursors in cells with or without crosslinker and analyzing eluates by immunoblotting for the presence of SpA_ED_ and SecA, identified the former but not the latter (see revised Figure 4).

Second, can one study a model substrate (precursor) that gets stuck in the membrane? Beckwith and colleagues studied MalE precursor fused β-galactosidase and demonstrated that the hybrid, once initiated into the Sec pathway, got stuck in the membrane, blocking the secretion of other precursors. The Beckwith lab used MalE-LacZ toxicity for genetic analyses of the secretory pathway; we suppose they could have also used a biochemical approach to isolate secretion machinery components alongside the arrested substrate. This was the experimental approach by Gottfried Schatz and colleagues, utilizing an impassable substrate (DHFR hybrids) to isolate components of the mitochondrial protein import machinery.

We elected for a different experimental path: isolating signal peptide mutants that are defective for precursor secretion. Emr and Silhavy used this approach to identify mutations in the coding sequence for the signal peptide of *E. coli lamB*; LamB is an outer membrane porin responsible for diffusion of maltodextrins into the bacterial periplasm. Similar to Silhavy and colleagues, we identified point mutants in the signal peptide of SpA_ED_ that were arrested in secretion. As we cannot apply selection to isolate genetic suppressors in *S. aureus*, we used the biochemical approach of Schatz and colleagues to isolate secretion machinery components that co-purified with the SpA_ED/S18L_ precursor. Of course, the reviewer is correct with his inquiry that interactions with the mutant signal peptide should be stable, i.e. they should not require a crosslinking reagent. To test this, we performed the affinity chromatography experiment with and without crosslinker. Following affinity chromatography of SpA_ED/S18L_ precursor and analysis of eluates via immunoblotting, we observed in the revised Figure 4 that SecA co-purifies with SpA_ED/S18L_ both in the presence and in the absence of the crosslinking agent.

Why were only SecA, SecDF and LtaS chosen for further analysis? We analyzed the mass spectrometry data for candidates that we presumed were likely to be involved in the secretion of SpA_ED_ either because they are known components of the protein secretion pathway (SecA/YEG/DF) or because they are located within septal membranes. In addition to LtaS, we analyzed also EzrA, PBP2, ClpB, ClpC, SAOUHSC_01854 (unknown function), SAOUHSC_02423 (UDP-N-acetylglucosamine pyrophosphorylase), SAOUHSC_01583 (conserved hypothetical phage protein), PurL, ClpB, PknB (serine/threonine protein kinase), RseB, and SAOUHSC_02406. However, in addition to *secA* and *secDF*, the *ltaS* depletion strain was the only mutant strain that exhibited a phenotype in septal secretion of SecA. We added a supplementary figure (Figure 4—figure supplement 1) and edited the text to illustrate our experimental plans and reasoning.

We did not use bacterial two hybrid studies to ask whether or not there are direct interactions between SpA_ED/S18L_ precursor and SecA, SecDF or LtaS. SecA is known to bind to the signal peptides of precursor proteins; this will likely include the SpA precursor. SecDF acts on unfolded polypeptides emerging from the secretory pathway (SecYEG), not on folded proteins purified by affinity chromatography. Our mass spectrometry analysis detected small amounts of SecDF, likely because SpA_ED/S18L_ associates not only with SecA but also with SecYEG/DF. We presume that LtaS does not directly interact with SpA_ED/S18L_. SpA_ED/S18L_ complexed with SecA may associate with LtaS following DDM solubilization of septal membranes. We modified the manuscript to make these points clear.

I am puzzled by the data presented in Figure 7 and 8: on one hand, SecA is everywhere (Figure 7), but upon depletion (albeit the conditions used here are unclear) SpA becomes diffuse (presumably in the membrane?) (Figure 8). Since the cells don't look sick (as in Figure 6D), this must be a mild SecA depletion. So the residual SecA is presumably everywhere in the membrane, but is not interacting with SpA at the septum? Why not? What provides the specificity for this localized interaction? Why does YSIRK/GXXS interact with only a subset of SecA molecules at the septum when SecA is everywhere? This key question remains unclear to me.

On the SecA observations, as is discussed in our first response to reviewer #2, SecA function is required for *E. coli* or *S. aureus* growth, but not SecA protein. Thus, after 3 hours of depletion, SecA is no longer detected yet SecA-depleted cells have not yet ceased growth. Nevertheless, SecA activity is diminished after 3 hours of depletion. After 6 hours of SecA-depletion, as secreted products are being degraded and cannot be replenished, *S. aureus* WY223 ceases growth, cells become bloated and eventually die. We show that in *S. aureus*, SecA associates with all membranes, i.e. SecA-GFP is not restricted to septal membranes, as microscopic localization studies reveal the presence of the protein in the vicinity of all membranes. However, Figure 8 shows that SpA, which requires SecA for secretion, can only be secreted at septal membranes by cells with active SecA. We infer from these data that SpA precursor (SpA_ED/S18L_) associates with SecA, however SecA-SpA complexes are only translocated at septal membranes, presumably because these membranes harbor a protease that removes the YSIRK-GXXS peptide, thereby allowing processed SpA to advance along the secretory pathway. What chemical feature defines septal membranes? We propose the production of lipoteichoic acid is the key chemical feature marking septal membranes for YSIRK/GXXS precursor translocation. In the absence of SecA, SpA precursors associate with all membranes, however they cannot traffic to septal membranes and be translocated at this site because they cannot enter the secretory pathway.